# $\text{S}^3$: Sign-Sparse-Shift Reparametrization for Effective Training of Low-bit Shift Networks

**Xinlin Li[1], Bang Liu[2], Yaoliang Yu[3], Wulong Liu[1], Chunjing Xu[1], and Vahid Partovi Nia[1]**

[1]Noah's Ark Lab, Huawei Technologies.
[2]Department of Computer Science and Operations Research (DIRO), University of Montreal.
[3]Cheriton School of Computer Science, University of Waterloo.

## Abstract

Shift neural networks reduce computation complexity by removing expensive multiplication operations and quantizing continuous weights into low-bit discrete values, which are fast and energy-efficient compared to conventional neural networks. However, existing shift networks are sensitive to the weight initialization and yield a degraded performance caused by vanishing gradient and weight sign freezing problem. To address these issues, we propose $\text{S}^3$ re-parameterization, a novel technique for training low-bit shift networks. Our method decomposes a discrete parameter in a sign-sparse-shift 3-fold manner. This way, it efficiently learns a low-bit network with weight dynamics similar to full-precision networks and insensitive to weight initialization. Our proposed training method pushes the boundaries of shift neural networks and shows 3-bit shift networks compete with their full-precision counterparts in terms of top-1 accuracy on ImageNet.

## 1 Introduction

While deep neural networks (DNNs) have achieved widely success in various tasks, the training and inference of DNNs usually cost prohibitive resources due to the fact that they are often over-parametrized and composed of computational costly multiplication operations for both forward and backward propagation [28]. To enable the application feasibility of DNNs in resource-constrained scenarios, substantial efforts have been made to reduce the computation complexity of neural networks while preserving their accuracy. Among different proposed approaches for this purpose, low-bit neural networks with binary weights [6, 22] or ternary weights [15] are recently designed to replace the expensive operations like multiplication with cheaper ones, e.g., replacing multiplication with sign flip operation during inference.

We focus on low-bit shift neural networks [11, 9, 30] that replace multiplication with the bit-shift operation. Following the conventional linear algebra notations, we denote scalars by $x$, vectors in bold small letters $\mathbf{x}$, and matrices by bold capital letters $\mathbf{X}$. Traditional neural networks require computing the inner product $\mathbf{w}^\top \mathbf{x} = \sum_i w_i x_i$, where $\mathbf{w}$ is the weight vector and $\mathbf{x}$ is the feature vector. In comparison, shift neural networks limit the values of the weight vectors to a set of discrete values $\mathbf{w}_{\text{shift}} \in \{0\} \cup \{\pm 2^p\}$. In this case, multiplication can be replaced with bit-shift operations, as multiplying $w_i = 2^p$ is equavilent to shifting $p$ places to the left (if $p$ is positive) or to the right (if $p$ is negative). The bit-shift operation alone only covers discrete neural networks with positive weights, therefore, most shift neural networks further add a sign flip operation to the shift to allow $w_i = \{\pm 2^p\}$. Inference with bit-shift operations is highly hardware friendly and can save up to $196\times$ energy cost on FPGA and $24\times$ on ASIC compare to their multiplication counterparts [28].

Several works aim to further improve the memory efficiency of shift neural networks by reducing the bit-width of the weights. Zhou et al. [30] propose to fine-tune pre-trained full-precision weights

with a progressive power-of-two weight quantization strategy. They suggest to keep a portion of weights unquantized during fine-tuning to improve model performance. DeepShift [9] is the state-of-the-art technique for training low-bit shift networks from random initialization. Initialization with pre-trained full-precision weights provides a significant performance gain for DeepShift. All latest techniques heavily rely on the initialization of full-precision pre-trained weights, which implies some optimization difficulties exist in the current training diagram of low-bit shift networks. However, although extremely memory efficient and hardware friendly, low-bit shift networks suffer from a significant accuracy drop on large datasets compared to their full-precision counterpart, and the performance of them is sensitive to weight initialization.

To address these problems, in this work, we analyze the current training diagram of low-bit shift neural networks, and find that the accuracy degradation and the weight initialization sensitivity of the low-bit shift networks are caused by the design flaw of the weight quantizer during training. Specifically, training shift neural networks is different from training full-precision networks. The gradient-based optimization algorithms are designed for optimizing continuous variables, but the weights of shift neural networks are discrete values. To learn the discrete weights with a gradient-based approach, it is necessary to exploit weight re-parameterization. One of the most widely adopted discrete weight re-parameterization technique is using a quantizer function to map a continuous parameter $w \in \mathbb{R}$ to a discrete weight $w \in \{v_1, v_2, \cdots, v_k\}$ with $k$ possible values. As the weight quantizers are generally non-differentiable functions, a gradient approximator, such as straight-through estimator [3], is utilized to approximate their derivatives during training. Our analysis shows the quantizer leads to a severe gradient vanishing and weight sign freezing.

In this paper, we design an effective training strategy to combat the gradient vanishing and weight sign freezing problem in low-bit shift networks. We first impose a dense weight priori during training, which equals maximizing the $\ell_0$ norm of the discrete weights. However, optimizing $\ell_0$ norm during training is a combinatorial problem, instead we propose $S^3$ re-parameterization and regularize the sparsity parameter. $S^3$ is a new technique which re-parameterizes the discrete weights of shift networks in a sign-sparse-shift, a 3-dimensional augmented parameter space to disentangle the roles of quantization values, and hopefully train more effectively thanks to more orthogonal axes in the augmented space. Although our method introduces extra parameters during training, its memory occupation under low-bit is slightly higher than the full-precision network since the additional memory overhead depends on the weight size. In contrast, the activation size dominates the training memory occupation, especially using a large batch size.

We evaluate our approach on the ImageNet dataset. While all previous methods require at least 5-bit weight representation to achieve the same performance as the full-precision neural networks on large datasets such as ImageNet, our experimental results show that our proposed method surpasses all previous methods, pushes this boundary further to 3-bits. Besides, our approach requires no complicated weight initialization or training strategy. Moreover, we define two indices of weight dynamics, named weight sign variation rate (WSVR) and weight low-value rate (WLVR), and compare the trend of these two indices during the training process of different methods. Experiments show that the weight dynamics of the ternary weights trained with our proposed technique better align with the weight dynamics of full-precision weights than trained with a traditional quantizers. This desirable dynamics is caused by our re-parametrization, and is the key to efficient training. We hope that this opens the possibility for methods similar as $S^3$ to be used in training low-bit networks.

## 2 Related Works

Different approaches have been proposed to reduce the computational complexity of neural networks. A type of the popular multiplication-free neural networks is low-bit neural networks with binary weights [6, 22] or ternary weights [15]. However, they suffer from under-fitting on large datasets, leading to an accuracy gap compared with their full-precision counterparts. Other approaches include replacing multiplication operations with addition operations [5, 25, 26] or bit-shift operations [11, 9, 30]. Although these approaches achieve a low accuracy drop on large datasets, they require higher weight representation bit-width. Other follow-up works try to improve the accuracy of multiplication-free neural networks by replacing multiplication with both addition and bit-shift operation [28], sum of binary bases [17, 29], or sum of shift kernels [16]. However, their computational cost is high as they use multiple operations per kernel.

Training shift neural networks with discrete weight values is a challenging task. Many weight re-parameterization approaches have been proposed to solve the challenge. One way is using stochastic weight [6, 18, 23], but these methods suffer from the slow computation of sampling. Another way is utilizing a quantizer function [6, 22, 15] to map or threshold continuous weights to discrete values. Since the thresholding function is non-differentiable, the backward gradients across the quantizer are approximated by a gradient approximator. The Straight-Through-Estimator (STE) [3] is a popular choice for quantizer gradient approximation.

There are also works study the vanishing gradient problem (VGP) that appears frequently in deep neural network training. [1, 10] point out that the zero values in full-precision weights lead to VGP when training highly pruned neural networks, which eventually jeopardizes the accuracy of the model. [8, 30] suggest to maintain a portion of the full-precision weights during quantized network training for improving the performance of the model. [30] shows that applying progressive quantization on low and high value weights differently can further improve the performance of the trained model. Quantizing high-value weights first and keep more low-value weights remains in full-precision during training helps to achieve a better shift neural network compared with randomly applying progressive quantization strategy on weights regardless their values. This observation coincide with our analysis, and we argue that the performance gain of progressive quantization methods come from the gradient flow preserved by maintaining a portion of full-precision weight during training. However, the study of the negative impacts caused by weight quantizers are still missing. To the best of our knowledge, we are the first to study the VGP caused by weight quantizer design.

## 3 Training Low-bit Networks

We first introduce the general form of the weight quantizers for discrete weight training, and then explain how quantization function leads to training difficulties. In low-bit networks, including shift networks, the weights are discrete values while usually the gradient-based optimization algorithms are designed for optimizing continuous variables. Specifically, quantizer function is one of the most widely-used approach for training discrete weight neural networks. It maps a continuous weight $w$ to one of the $k$ discrete weight values $Q(w) \in \{v_1, v_2, \cdots, v_k\}$. Several quantizers of low-bit networks [15, 31, 27] and shift networks [9, 30] are in the following form of staircase function:

$$Q(w) = \begin{cases} v_1 & w < t_1 \\ v_2 & t_1 \leq w < t_2 \\ \vdots \\ v_k & t_k \leq w \end{cases} \tag{1}$$

However, training a quantized neural network with a quantizer function in the form of Equation (1) is challenging due to two issues, vanishing gradient (VGP) and weight sign freezing. We discuss the vanishing gradient issue in section 3.1 and the weight sign freezing issue in section 3.2.

### 3.1 Gradient vanishing

To clarify further why quantization and VGP are entangled, suppose a ReLU-activated fully-connected layer can be generally formulated as:

$$\begin{bmatrix} x_0^{l+1} ' \\ x_1^{l+1} \end{bmatrix} = \text{ReLU}(\begin{bmatrix} W_{00}^l & W_{01}^l \\ W_{10}^l & W_{11}^l \end{bmatrix} \begin{bmatrix} x_0^l \\ x_1^l \end{bmatrix}) \tag{2}$$

The gradient update toward $x_0^l$ can be calculated as:

$$\frac{\partial \text{Loss}}{\partial x_0^l} = \frac{\partial \text{Loss}}{\partial x_0^{l+1}} \frac{\partial x_0^{l+1}}{\partial (W_{00}^l x_0^l + W_{01}^l x_1^l)} W_{00}^l + \frac{\partial \text{Loss}}{\partial x_1^{l+1}} \underbrace{\frac{\partial x_1^{l+1}}{\partial (W_{10}^l x_0^l + W_{11}^l x_1^l)}}_{\text{Derivative of ReLU}} W_{10}^l \tag{3}$$

And the derivative of ReLU is,

$$\frac{\partial x_1^{l+1}}{\partial (W_{10}^l x_0^l + W_{11}^l x_1^l)} = \begin{cases} 1 & W_{10}^l x_0^l + W_{11}^l x_1^l > 0 \\ 0 & W_{10}^l x_0^l + W_{11}^l x_1^l \leq 0 \end{cases} \tag{4}$$

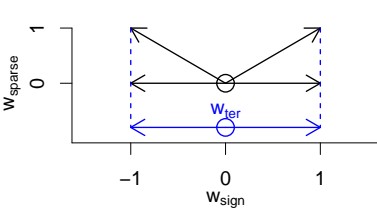
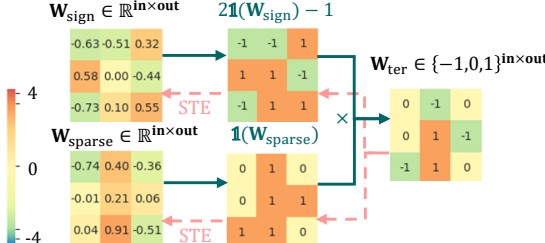

Figure 1: Left panel: decomposing $w_{\text{ter}}$ in Equation (6). Blue vectors demonstrate the original ternary space, while the black vectors show the decomposed augmented space. The augmented parameters are projected back to the ternary space after multiplication . Right panel: efficient back propagation on $w_{\text{ter}}$ in the $S^3$ re-parameterized space.

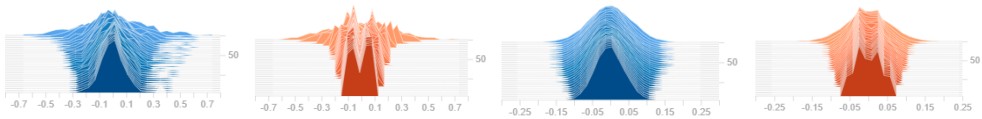

| (a) 3-th layer | (b) 3-th layer, ternary | (c) 7-th layer. | (d) 7-th layer, ternary |

Figure 2: Density polygon of the weights in the multiple layers of ResNet20 trained on CIFAR-10, full-precision (blue), versus ternary before passing through the quantization function (orange). Full-precision and quantized training have different dynamics. Ternary weights are kept away from zero, preventing weight sign change.

By observing Equation (3), we can see that the gradient information $\frac{\partial \text{Loss}}{\partial x_1^{l+1}}$ propagating from $x_1^{l+1}$ to $x_0^l$ vanishes in two distinct situations: i) the corresponding derivative of ReLU is zero. (i.e. the neuron is not activated), and ii) the corresponding weight $W_{10}^l$ is zero. Both scenarios lead to a gradient information loss, and ultimately cause inferior performance even after training convergence. The situation i) is widely known as the Dying ReLU issue caused by the sparse derivative of ReLU. Several ReLU-variants [20, 12] with non-sparse derivatives are developed as remedies.

The VGP caused by the quantizer is similar to the dying ReLU issue. As the first step, we motivate $S^3$ technique on a ternary network and then generalize it further for higher bit shift networks. The ternary quantizer function (1) contributes to situation ii) by mapping a portion of non-zero full-precision weights $w$ to the exact zero value $w_{\text{ter}} = Q(w)$. This leads to a smaller $\ell_0$ norm for the discrete weight term $w_{\text{ter}}$ compared to its full-precision counterpart $w$, $\|Q(\mathbf{w})\|_0 \leq \|\mathbf{w}\|_0$. Shift networks exchange the costly multiplication operation with cheap shift operation, so inherently it is a non-uniform quantized network and faces the same problem. The VGP caused by the quantizer is the main optimization difficulty in training shift neural networks as well. This issue appears more severely in low-bit networks which include fewer quantized range with higher sparsity rate.

## 3.2 Weight sign freezing

Another issue of the low-bit quantizer is the weight sign freezing effect. This effect prevents the weight sign variation and leads to different weight dynamics between the full-precision weights and the low-bit discrete weights.

The weight dynamics of full-precision weights are as follows. In general practice, the weights are initialized with a zero-centred distribution in small variance and trained with $\ell_2$ regularization, so they stay close to the origin point during training. When the full-precision neural network train with noisy gradient updates, such as an SGD optimizer and a large learning rate at the beginning of training, the weights oscillate around the origin point. With the learning rate decreasing, more and more weights converge to a specific sign representing the corresponding input feature, which correlates to the prediction output. The remaining weights whose input feature is neither positively correlated nor negatively correlated are concentrated in the region close to the origin point to oscillate.

However, we notice that the weight dynamics of low-bit discrete weights trained with quantizer are dramatically different. We train ResNet-20 models with full precision weights and ternary weights using quantizer and compare the histogram variation of the same layer in Fig. 2. Take ternary quantizer [15] as an example. The histogram variation of a ternary weight trained with the quantizer has two peaks. They are symmetric to zero; this is because the positive and negative thresholds of the ternary quantizer attract a large portion of weights oscillating around them and prevent the weights across zero and switching their sign. The intuition of full-precision weight dynamics and the observation from weight histogram comparisons inspire us to design the experiments in section 5.2. Our experiments show that the ternary networks trained with a quantizer have very low WSVR during the whole training process. We call it the weight sign freezing issue.

# 4 $S^3$ Re-parameterization

We suggest to initialize the networks with **dense weight prior**, which is equivalent to **maximizing the $\ell_0$ norm of the weight tensors at initialization**. We initialize ternary weights away from zero so that VGP is partially solved at early training epochs. Dense weight initialization provide additional benefits for low-capacity models, and encourages training around non-zero weight values. This characteristic empirically increases the model capacity and alleviate under-fitting problem commonly observed in low-bit models.

The remedy proposed at initialization only solves training difficulty at early training. Efficient training over discrete weights still remain slow. As the follow up remedy we propose to re-parameterize the discrete weight of shift networks into a sign-sparse-shift 3-fold manner, we call $S^3$. This re-parameterization allows us to regard the low bit network as projection of an augmented space, in which a single parameter takes care sparsity, a single parameter takes care of of switching signs, and another parameters takes care of the weight magnitude. This decomposition has two benefits. First, it promote the weight sign variation by allowing the discrete weight switch between positive and negative values without approaching to zero. Second, it allows to control the amount of sparsity by controlling the sparsity parameter during longer epochs.

## 4.1 Ternary network

Almost all ternary networks such as [15] are trained with the following quantizer

$$w_{\text{ter}} = Q_\Delta(w) = \begin{cases} 1 & \Delta \leq w, \\ 0 & -\Delta \leq w < \Delta, \\ -1 & w < -\Delta. \end{cases} \tag{5}$$

We propose to replace the quantizer in the form of (5) with the following $S^3$ re-parameterized counterpart, see Figure 1:

$$w_{\text{ter}} = \mathbb{1}(w_{\text{sparse}})\{2\mathbb{1}(w_{\text{sign}}) - 1\}, \tag{6}$$

in which $\mathbb{1}(\cdot)$ is the Heaviside function, i.e. $\mathbb{1}(x) = 1$ if $x > 0$ and equals zero otherwise. Note that $w, w_{\text{sparse}}, w_{\text{sign}}$ are full precision parameters, and ternary weights are recovered after passing through the Heavidside function.

The $S^3$ re-parameterization approach for ternary weight training decomposes the discrete ternary weight into two binary parameters, one representing sign, another representing sparsity. Motivated from the ReLU activation, we propose to impose dense weights by penalizing the loss function with the magnitude of negative weights. This is a simple regularization that adds little to training computation and is effective to enforce dense weights.

$$\mathcal{R}_{\text{sparse}}(\mathbf{w}_{\text{sparse}}) = \|\max(-\mathbf{w}_{\text{sparse}}, 0)\|_1 \tag{7}$$

## 4.2 Shift networks

We further generalize our approach toward shift neural networks. Following [9] the discrete weight of shift networks can be decomposed into two parts: a ternary term $w_{\text{ter}}$ and a scaling term composed of power of 2 numbers $2^S$ that shifts the network $S$ bits $w_{\text{shift}} = w_{\text{ter}}2^S$, shifting to left or right depends on the sign of $S \in \mathbb{Z}$.

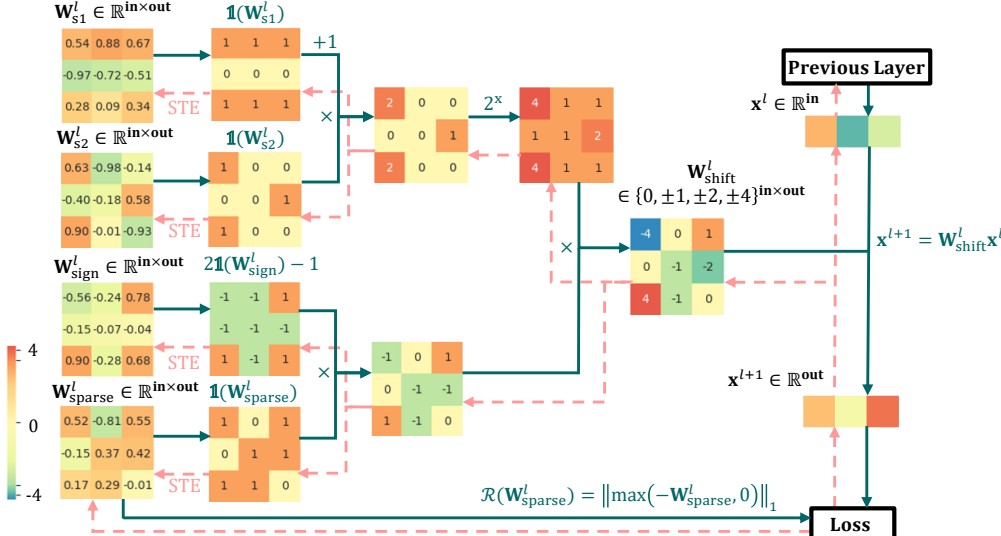

Figure 3: Illustrating the overview of 3bit shift network trained with $S^3$ re-parameterization and dense weight regularizer (forward computation in dark green, backward in light red)

We apply the decomposed $w_{\text{ter}} = w_{\text{sign}} w_{\text{sparse}}$ described in the session 4.1 and re-write shift networks in terms of the decomposed ternary weights. Following [9] we focus on positive $S$ values. The re-parameterization for the negative $S$ values can easily be achieved by adding a constant bias to $S$. We propose to re-parameterize this shift weights as a combination of $t$ binary variables recursively.

$$S_0 = 0, S_t = \mathbb{1}(w_t)(S_{t-1} + 1). \tag{8}$$

For instance, $S^3$ reparameterizes a 3-bit shift networks whose discrete weight values are $w_{\text{shift}} \in \{0, \pm 1, \pm 2, \pm 4\}$ to:

$$w_{\text{shift}} = 2^{S_2} \mathbb{1}(w_{\text{sparse}})\{2\mathbb{1}(w_{\text{sign}}) - 1\}, \tag{9}$$

while $S_2 = \mathbb{1}(w_2)\{\mathbb{1}(w_1) + 1\}$, so it reduces to a ternary network for $t = 0$. Note that in this reparametrization all weights $(w_{\text{sign}}, w_{\text{sparse}}, w_1, w_2)$ are trained in full precision. Figure 3 demonstrates forward pass and back propagation using this reparametrization.

Ultimately

$$\mathcal{L}(\mathbf{w}) = \text{Loss}(\mathbf{w}) + \lambda \mathcal{R}(\mathbf{w}) + \alpha \mathcal{R}_{\text{sparse}}(\mathbf{w})$$

is optimized, in which $\mathcal{R}$ is the $\ell_2$ norm and $\alpha > 0$ is a regularization constant controlling the network sparsity during training.

## 5 Experiments

In this section, we describe our experiment setup and benchmark $S^3$ re-parameterization over SOTA low-bit DNNs. Then we compare the dynamics of weights with different training paradigms to show our method could better align with the training of full-precision models. Finally, we present ablation studies of the dense weight regularizer and the number of training epochs.

### 5.1 Benchmark $S^3$ re-parameterization over SOTA low-bit DNNs

**Models and datasets.** We evaluate our proposed method on ILSVRC2012 [7] dataset with different bit-widths to demonstrate the effectiveness and robustness of our method. We use ResNet-18 and ResNet-50 as our backbone with the same data augmentation and pre-processing strategy proposed in [13]. Following common practice in most previous methods [22, 19, 29], all convolution layers are quantized except for the first one.

Table 1: Comparison of SOTA methods using ResNet-18 trained on ImageNet

| Kernel operation | Methods | Bit-Width | Initialization | Top-1 Acc. (%) | Top-5 Acc. (%) |
|---|---|---|---|---|---|
| Multiplication | FP32 | 32 | Random | 69.6 | 89.2 |
| | TTQ | 2 | Pre-trained | 66.6 | 87.2 |
| Sum of Sign Flips | Lq-Net [29] | 2 | Random | 68.0 | 88.0 |
| | Lq-Net [29] | 3 | Random | 69.3 | 88.8 |
| | Lq-Net [29] | 4 | Random | 70.0 | 89.1 |
| Sign Flip | BWN [22] | 1 | Random | 60.8 | 83.0 |
| | HWGQ [4] | 1 | Random | 61.3 | 83.2 |
| | BWHN [14] | 1 | Random | 64.3 | 85.9 |
| | IR-net [21] | 1 | Random | 66.5 | 86.8 |
| | TWN [15] | 2 | Random | 61.8 | 84.2 |
| | LR-net [23] | 2 | Random | 63.5 | 84.8 |
| | SQ-TWN [8] | 2 | Random | 63.8 | 85.7 |
| | INQ [30] | 2 | Pre-trained | 66.02 | 87.13 |
| | Ours | **2** | Random | **66.37** | **87.18** |
| Shift + Sign Flip | INQ [30] | 3 | Pre-trained | 68.08 | 88.36 |
| | INQ [30] | 4 | Pre-trained | 68.89 | 89.01 |
| | INQ [30] | 5 | Pre-trained | 68.98 | 89.10 |
| | DeepShift [9] | 6 | Random | 65.63 | 86.33 |
| | DeepShift [9] | 6 | Pre-trained | 68.32 | 88.41 |
| | Ours | **3** | Random | **69.82** | **89.23** |
| | Ours | **4** | Random | **70.47** | **89.93** |

Table 2: Comparison of SOTA methods using ResNet-50 trained on ImageNet

| Kernel operation | Methods | Bit-Width | Initialization | Top-1 Acc. (%) | Top-1 Acc. (%) |
|---|---|---|---|---|---|
| Multiplication | FP32 | 32 | Random | 76.00 | 93.00 |
| Shift + Sign Flip | INQ [30] | 5 | Pre-trained | 74.81 | 92.45 |
| | DeepShift [9] | 6 | Pre-trained | 75.29 | 92.55 |
| | Ours | **3** | Random | **75.75** | **92.80** |

**Training settings.** We train the networks with 200 epochs utilizing the cosine learning rate, and the initial learning rate is 1e-3. The networks are optimized with SGD optimizer, and the momentum and weight decay are set to 0.9 and 1e-4 respectively. The hyper-parameter $\alpha$ of dense weight regularizer is set to 1e-5 without using decay scheduler.

**Baselines.** We evaluate the proposed approach over two SOTA methods for training low-bit shift networks, including DeepShift [9] and Incremental Network Quantization (INQ) [30]. We also compare the results of shift network to other neural networks with higher computational cost, including multiplication-based ConvNet in full-precision [13] and LqNet [29].

**Experiment results.** The results summarized in Table 1 and Table 2 show that our method significantly improves the accuracies of low-bit multiplication-free networks. Our proposed method on the 3-bit shift networks has the most exciting result: it achieves state-of-the-art accuracy of low-bit shift networks with a Top-1 accuracy of 69.82% on ResNet-18 and 75.75% on ResNet-50 without extra technique at initialization or during training.

## 5.2 Compare and analyze the dynamics of weights with different training paradigms

The dynamics of weights during training reveals the learning process of a neural network. Here we define two indices reflecting the weight variation behaviour during training, namely weight sign variation rate (WSVR) and weight low-value rate (WLVR). Then we compare the variation trend of them among different models during training.

**Weight Sign Variation Rate** (WSVR) indicates the frequency of weight sign variation during training. To measure WSVR, we take weight snapshots every 10 epochs during training and measure the percentage of weights with a different sign between two neighbouring snapshots as the WSVR.

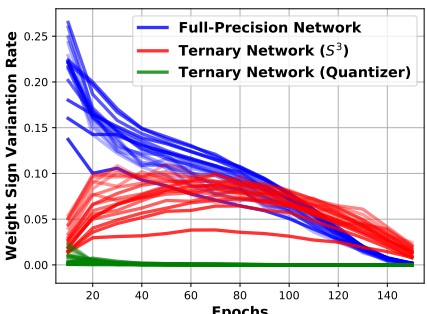 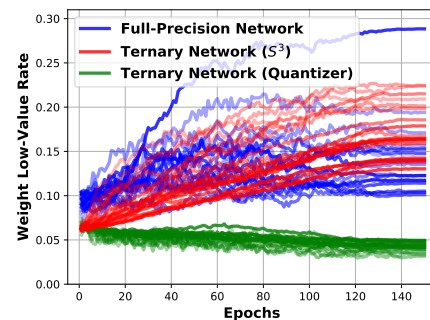

(a) Weight Sign Variation Rate        (b) Weight Low-Value Rate

Figure 4: Weight dynamics comparison using ResNet-20 trained on CIFAR-10. Each curve represents one layer in the model. The colour of the layer gradually becomes opaque from front to end.

**Weight Low-Value Rate** (WLVR) indicates the percentage of weight values relatively close or equal to zero. We use it as a unified index of continuous and discrete weights to reflect their generalized sparseness during training. For discrete weight networks, we consider weight sparsity as WLVR. For a model with continuous weight trained without an $\ell_1$ regularizer, the low-value weights oscillate in a neighbouring region of zero. To measure WLVR, we normalize the weight tensors of each layer to the range between -1 and 1, then count the percentage of weights between -0.03 and 0.03 as WLVR. Our conclusion relies on the variation trend of the curves, which is not significantly affected by this hyper-parameter.

**Experiments setup.** We compare the two indicators mentioned above on three models: a full-precision model, a ternary weight model including quantizer [15], and an $S^3$ re-parameterized ternary weight model described in section 4.1. This analysis chooses the ResNet-20 model as the backbone and trains all three models on the CIFAR-10 dataset for 150 epochs with a cosine annealing learning rate and the initial learning rate set to 0.1. All models are converged and reach a reasonable validation accuracy on CIFAR10 ($> 91\%$). We measure WSVRs and WLVRs for each layer in all three models during training and summarize the results in Figure 4.

**Result analysis.** We can observe from Figure 4a that the full-precision network has high WSVRs at the beginning of training and then slowly decreases to zero in the end, indicating that the weight signs oscillate rapidly initially and gradually stabilize in the end. On the contrary, the ternary weight model with a traditional quantizer remains low WSVRs during the whole training process, indicating that the weight sign variation is not as frequent as its full-precision counterpart. Although our proposed approach has a different trend of WSVRs with the full-precision model initially, it is in good alignment in the rest of the training stage.

Figure 4b shows that the full-precision network has low WLVRs in the early stage of training and then slowly increases, indicating that the portion of low-value weights increases during the training of the full precision model. However, the WLVRs variation trend of the ternary weight network with quantizer is toward the opposite direction—the weight sparsity of the ternary weight network trained with a quantizer decrease during training. On the trend of this indicator, our method better aligns with the full-precision model.

We argue that the design of $S^3$ re-parameterization is the key to better alignment with the weight dynamics. By decomposing into $w_{sign}$ and $w_{sparse}$, the discrete weight can oscillate between -1 to 1 values directly, and the dense weight regularizer keeps the $w_{sparse}$ stay at one as long as possible until both +1 and -1 status increasing the loss value and the gradient signal from the optimizer push the discrete weight values to zero status. This decomposition design leads to weight dynamics very close to the full-precision weight dynamics described in section 3.2, i.e. oscillation and slowly converge to a specific weight sign, otherwise push to zero.

### 5.3 Ablation studies

In this section, we verify the efficiency and robustness of our method via extensive experiments.

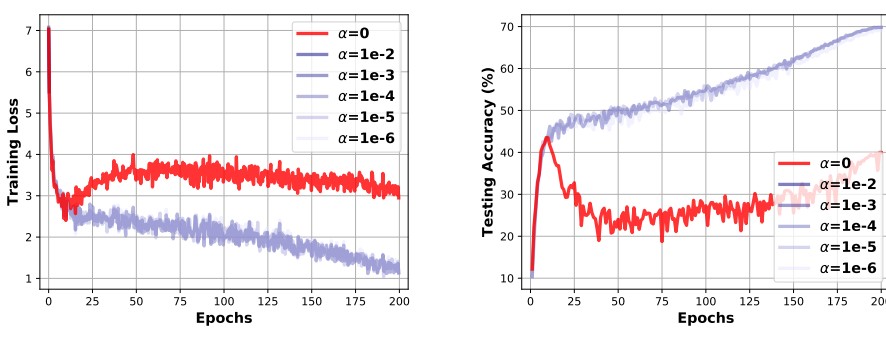

(a) Training Loss          (b) Testing Accuracy

Figure 5: Sensitivity analysis for dense weight regularizers without using decay scheduler

Table 3: Dense weight regularizer hyper-parameter comparisons using ResNet-18 on ImageNet

| Decay Scheduler | ResNet-18 ImageNet | $\alpha$ | | | | |
|---|---|---|---|---|---|---|
| | | 1e-2 | 1e-3 | 1e-4 | 1e-5 | 1e-6 |
| None | Top-1 | 69.78 | 69.85 | 69.84 | 69.82 | 69.83 |
| | Top-5 | 89.21 | 89.24 | 89.23 | 89.23 | 89.23 |
| Linear | Top-1 | **69.91** | 69.68 | 69.74 | 69.62 | 66.70 |
| | Top-5 | **89.23** | 89.06 | 89.13 | 89.10 | 86.93 |
| Cosine | Top-1 | 69.91 | 69.85 | 69.85 | 69.57 | 65.82 |
| | Top-5 | 89.15 | 89.20 | 89.38 | 89.04 | 86.50 |

**Dense weight regularizer.** We verify a wide range of hyper-parameters of the dense weight regularizer and two decay schedulers: linear decay and cosine decay, to prove our method's efficiency and robustness. We choose the 3-bit $S^3$ re-parameterized shift network and ResNet-18 backbone on the ImageNet dataset as the test case. All hyper-parameters are the same as described in section 5.1. The ablation study results are summarized to Table 3.

The results show that the performance of $S^3$ re-parameterized networks is insensitive to the choice of $\alpha$ value of the dense weight regularizer. The decay schedulers for $\alpha$ have a limited impact on the performance of the trained model as well, indicating that the main benefit of dense weight priori comes from the early stage of training. This result implies the performance gain from initialization with a pre-trained full-precision model may come from the good initial estimation of weight sign from the pre-trained weights.

Although the experiment results are consistent across a wide range of $\alpha$ values and two different decay schedulers, Figure 5 shows that the dense weight regularizer is an indispensable part to train an $S^3$ re-parameterized network—otherwise, the training suffers from a convergence problem.

**Training epochs.** Due to their frequent sign variation instability, binary parameters' training requires more epochs compared to their full-precision counterparts [6, 24, 2]. Since the number of training epochs is critical for binary parameter training, we verify the neural network performance affected by it. We choose the 3-bit $S^3$ re-parameterized shift network and ResNet-18/50 backbone on ImageNet dataset as the test case. All hyper-parameters are the same as described in section 5.1, except for the number of epochs. Table 4 shows that the network performance can significantly improve with more epochs. With the training of 200 epochs, we close the accuracy gap between 3bit shift networks and full-precision CNN on ResNet18/50 ImageNet experiments.

## 6 Conclusion

Although low-bit shift networks are memory efficient and hardware friendly during inference, they are not able to achieve competing performance with their full-precision counterparts due to the gradient vanishing and weight sign freezing problems when trained by existing methods. Our proposed $S^3$ reparametrization technique efficiently addresses these issues and bridges the accuracy gap between

Table 4: Training epochs comparisons using ResNets trained on ImageNet

| Experiments | Accuracy | $\mathcal{R}_{\text{sparse}}$ $\alpha$ | Epochs | | | | |
|---|---|---|---|---|---|---|---|
| | | | 90 | 120 | 150 | 180 | 200 |
| ResNet-18 | Top-1 | 1e-5 | 68.45 | 69.24 | 69.28 | 69.63 | **69.83** |
| ImageNet | Top-5 | | 88.35 | 88.77 | 88.93 | 89.16 | **89.23** |
| ResNet-50 | Top-1 | 1e-5 | 74.67 | 75.47 | **75.76** | / | 75.75 |
| ImageNet | Top-5 | | 92.27 | 92.60 | 92.72 | / | **92.80** |

low-bit shift networks and their full-precision counterparts by carefully design a decomposed space for optimization so the discrete weight dynamics can match the continuous weight dynamics. Extensive experiments and ablation studies demonstrate the superior effectiveness and robustness of our method for training low-bit shift networks. Moreover, we show that $S^3$ reparametrization enables a better alignment between the dynamics of weights in training a low-bit shift network and its full-precision counterpart. Our future work is to further explore the theoretical ground of our reparametrization technique and weight dynamics.

## Acknowledgement

We would like to thank the area chair and the anonymous reviewers for their thoughtful discussions that led to significant improvement of the paper. We want to thank our colleagues Yunhe Wang and Chao Xing for providing us with helpful suggestions during the initial development phase of the paper. We also enjoyed our technical discussions with Mostafa Elhousie about DeepShift networks. We appreciate the initial hardware cost analysis by our colleague Seyed Alireza Ghaffari.

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
