# A  Appendix

## A.1  Comparison of the power-of-two scaling factor parameterizations

This experiment compares different parameterizations of the power-of-two scaling factors on a 3-bit shift network. Our proposed method describe in Eq.8, and another is a staircase-like quantizer function. Besides different scaling factor parameterization, all other hyper-parameters and designs are the same as the experiments in Table 3 except for training epochs reduced to 120.

The design of the staircase-like quantizer is following the general practice of quantization-aware training. In 3bit shift network, the value of power-of-two scaling factors is limited to $\{0, 1, 2\}$. During the forward propagation, shift parameters rescale to the range of (-0.5, 2.5) based on their min and max, and then rounded. During the backward propagation, we use STE to estimate the gradient across the staircase-like quantizer function.

Our experiment results summarized in Table 5, it shows that our proposed parameterization of the power-of-two scaling factor improves the shift networks' performance.

Table 5: Comparison of power-of-two scaling factor parameterizations using ResNet-18 on ImageNet

| Parameterization of $2^p$ | ResNet-18 ImageNet | $\alpha$ | | | |
|---|---|---|---|---|---|
| | | 1e-2 | 1e-3 | 1e-4 | 1e-5 |
| Staircase-like function | Top-1 | 68.58 | 68.58 | 68.52 | 68.59 |
| | Top-5 | 88.50 | 88.41 | 88.30 | 88.47 |
| Ours(S$^3$) | Top-1 | 69.78 | 69.85 | 69.84 | 69.82 |
| | Top-5 | 89.21 | 89.24 | 89.23 | 89.23 |

## A.2  Figures of the training epochs experiments

Figure 6 and 7 shows the training loss curves and testing accuracy curves of the training epochs experiments summarized in Table 4.

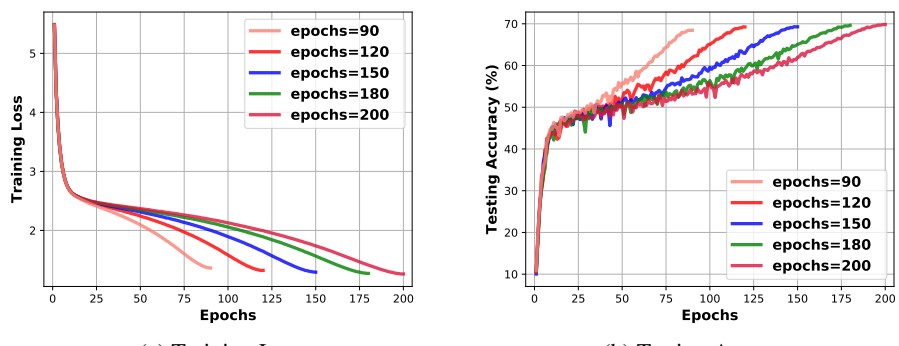

(a) Training Loss                    (b) Testing Accuracy

Figure 6: Training epochs comparison using ResNet-18 trained on ImageNet

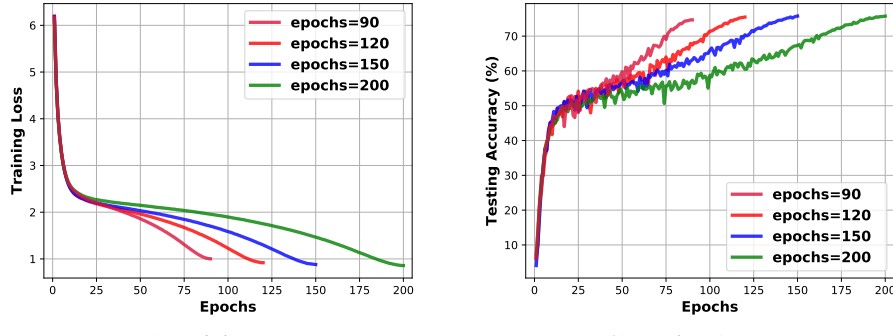

(a) Training Loss                                    (b) Testing Accuracy

Figure 7: Training epochs comparison using ResNet-50 trained on ImageNet