# OpenReview forum: "S$^3$: Sign-Sparse-Shift Reparametrization for Effective Training of Low-bit Shift Networks"
_NeurIPS.cc/2021/Conference — NeurIPS 2021 Poster_

### Official Review · Reviewer_KeAd · 2021-06-24

**Rating:** 2
**Confidence:** 4

**Summary:**

The paper proposes an idea of a different reparametrization for ternary networks, decomposing any values into learning a 'shadow weight' for both the sign and the value itself.



**Main Review:**

To me, this paper is a strong reject as it currently stands. The paper is written in a haphazard fashion, is imprecise in it's claims and methodology, and unconvincing in it's justifications of the method and the results. I will go over the parts of the paper that are unclear first from a methodological point of view.

Section 3.1 on the vanishing gradient problem is very imprecise. W^l being zero, and sparse, are not the same thing. I don’t see how a regularizer helps weights to be non-zero to counteract this. Also, sparse weights don’t necessarily lead to vanishing gradients; you could 0-out 50% of your weights, and double the remaining ones, and you’d still have a similar magnitude in your gradients. (1) refers to a quantiser, but not necessarily to a ternary one, this should be made more precise. The paper claims to be the first to analyze the vanishing gradient problem for quantization... but I don't see any further analysis of this that leaves me convinced this is a real issue. A plot to show vanishing gradients happening due to quantization would help.

Section 3.2 I don't see it proven that the sign is the actual issue here. If the same analysis was done for 4 bit quantization, wouldn't we see that the same problem occurs around +2 and +3? There doesn't seem to be any evidence for the sign itself being very specific to this problem. And if the sign isn't specifically the problem, I don't see how the suggested parametrization helps.

Section 4 - The underlying assumption that a weight that is 0, stays 0 and somehow ensures the network has a lower capacity doesn’t hold. Even when initialised at 0, weights can update to be higher or lower, unless full output neurons are initialised to be entirely 0 or you have a dying ReLU-like problem.

Discussing the reparametrization (4) in ternary, it seems to me that for w_ter = {w_sparse, w_sign}, that {0, 0} ->-1, {1,0}-> -1, {0, 1} -> 0, {1,1}->1. It is insightful to plot what happens with the values w_sparse and w_sign (e.g. wolfram alpha z = x * (2*y-1) for x=-0.1...1.1 and y=-0.1...1.1 ). In this curve, it's clear that if you initialize anywhere and want to move your value of z up (with a positive gradient), both w_sparse and w_sign would be increased. However, if the gradient of z has to go down, w_sign goes down but w_sparse goes up too! This means that when optimizing, w_sparse will always go up, and only w_sign is left as a factor that is actually optimized over meaningfully. This means that with this refactorization, you end up with one significant variable you are optimizing anyway. It would be insightful if the authors plotted the behavior of these gates for a toy example, and showed e.g. that jumps from -1 directly to 1 could happen with the shadow weights (I contest this is not possible in a smooth manner with the current formulation).

One very important point is that to me, the arguments for the weight initialization and the WSVR issue are at odds with each other. For weight initialization, it's mentioned that weights should be initialized far from 0, and the paper introduces the dense weight regularizer to actively encourage weights to be either -1 or 1. However, when discussing the weight sign freezing issue, the argument is that weights don't switch enough from -1 to 1. Weights that would be close to 0 would normally have an ability to switch to -1 or 1 when doing a normal optimization. We see that in graph 4.b, the ternary network (quantizer) remains on a very low sparsity, while the other networks gain higher sparsity over time. So despite the dense weight regularizer, the ternary network (S3) gets higher sparsity than the ternay network (quantizer)? That's the opposite of what the logical argument of the authors would suggest.

The results section is unclear to me. the 'sign flip' and 'shift+sign flip' methods are not introduced as is. It's also unclear where the difference in performance between methods is coming from. In the ResNet18 table, it seems the shift+sign flip method is better than the baseline. It might just be that the author's training pipeline is better? It should be clear how the other methods are ran for comparison, to see if this is fair. It's also unclear where the gains are coming from. Which part of the method is working? Is it the initialization? The quantization scheme? An ablation study that shows the difference between everything that is added would help tremendously. Perhaps INQ + better initialization could already give a better result?

The WSVR doesn’t seem like a sensible measure to me. I could have a procedure where I just randomly flip all weights in the network every epoch, which wouldn’t help with optimisation at all, but would give me a perfect WSVR score. If anything, 4.a. just shows to me that the learning rate is low for the ternary network, as weights not flipping means essentially nothing is being learned (as there’s just 3 possible values of the weights!).
There’s a more sensible analysis to show if your ternary optimisation method works, and it’s to actually see how your gradient estimate predicts a real decrease in the loss function. And the authors fail to show that in this paper.

From a structure point of view:
Please introduce shift networks, and your final method earlier in the text; all of section 3 talks about shift networks, but you haven’t introduced in your paper what you are talking about. The reader now has to refer to another work to figure out what shift networks are, and hope it’s the same notion as you mention. Please define what you mean by this yourself in the text.
Section 3.2 WSVR is named as an acronym for the first time since the introduction. Please properly name and define this before using it as an acronym.
“Different weight dynamics” What do you mean with this exactly, please be more precise.
Section 4.2 leaves me confused. First we’re looking at ternary networks, and S indicates we shift left or right. Then S is claimed to be positive, so I guess we only shift right? Then S2 is being introduced as ternary networks, but it seems the rest of the paper uses the S^3 definition as a ternary network (see fig 4.3)? It would really help if the authors worked on the flow of the article, clarifying what their actual method is and building up the understanding, and make sure there is consistency in the paper.
The order of figure 1 and 2 is reversed

**Time Spent Reviewing:**

5

---

> ### Author Response · Authors · 2021-08-10
> **Author response**
>
> Thank you very much for your detailed review. We summarize your concerns as follows.
>
> $\def\wshift{w_\mathrm{shift}}
> \def\wsparse{w_\mathrm{sparse}}
> \def\wsign{w_\mathrm{sign}}
> \def\wter{w_\mathrm{ter}}$
>
> #### **Q1:**  The flow of this paper is confusing and difficult to follow. Some terminologies are imprecise.
>
> #### **A1:** As all three other reviewers agree, this paper is **clearly written** (reviewer CUjS, FA7K, geT6) and **easy to follow** (reviewer CUjS), **the layout is good to read** (reviewer FA7K). We understand your concern and will use more precise language to avoid confusion in the camera-ready version.
>
>
>
> #### **Q2:** The insightful visualization (wolfram alpha z = x * (2*y-1) for x=-0.1...1.1 and y=-0.1...1.) shows the gradient update $\frac{\partial \text{Loss}}{\partial \wsparse}$ is always positive (go up) regardless of the sign of $\frac{\partial \text{Loss}}{\partial \wter}$.
>
> ####  **A2: The insight of the gradient update direction toward $\wsparse$ is wrong.** $\wsparse$ does not always go up. A counterexample is $\wsparse$ will go down when $\wter$ is negative ($\wsign<0$) and receive a positive gradient (want to go up).
> The gradient update for $\wsparse$ in the ternary case (eq.4) is derived as followed,
>
> $\wter =  \unicode{x1D7D9}(\wsparse) (2\unicode{x1D7D9}(\wsign)-1)$
> $\frac{\partial \text{Loss}}{\partial \wsparse}=\frac{\partial \text{Loss}}{\partial \wter}\frac{\partial \wter}{\partial \unicode{x1D7D9}(\wsparse)}\frac{\partial \unicode{x1D7D9}(\wsparse)}{\partial \wsparse}=\frac{\partial \text{Loss}}{\partial \wter}(2\unicode{x1D7D9}\wsign)-1)$
> Using STE, we assume,$\frac{\partial \unicode{x1D7D9}(\wsparse)}{\partial \wsparse}=1$
> In conclusion, the gradient update direction toward $\wsparse$ is depends on the binarized value of $\wsign$,$(2\unicode{x1D7D9}(\wsign)-1)\in(-1,1) $.
>
>
>
> #### **Q3:** The arguments in the original paper are at odds with each other. (1) Weight should be initialized far from 0 (2) Add a regularizer to encourage the weights to stay away from 0. (3) Weight should be encouraged to switch frequently between positive and negative. Argument (1) and (2) contradicts to argument (3) since the weight close to zero is easier to switch between positive and negative.
>
> #### **A3: We think that your understanding of arguments (1) and (2) is not accurate.** We propose to maximize the L0 norm of the weight tensor at initialization (line 161-162) and use a dense weight regularizer to encourage a large L0 norm of weight during training (eq 5); this is not equivalent to staying far from zero. Perhaps you confuse L0 norm with L1 norm. **A weight value can stay very close to 0. As long as it is not equal to 0, the L0 norm remains the same.**
>
> #### **Q4:** High WSVR (weight sign flipping rapidly) is not a sensible measure for the optimization process. Randomly weight sign flipping can lead to a high WSVR but will not lead to a lower loss value. The low WSVR of the ternary network shown in figure 4.a is due to a low learning rate during the training of the ternary network.
>
> #### **A4:**  First, **the learning rate is sufficiently high in our ternary network experiment.** Section 5.2 (line 249) mentions that all three models train with the same initial learning rate 0.1. Several ternary weight network papers suggest this learning rate to train their networks on ResNet-20 CIFAR10 [1,2,3].
>
> The CIFAR10 experiments are designed to demonstrate the working mechanism behind this method. A fair loss values comparison across re-parameterization methods is difficult to prepare on CIFAR10 since many low-bit networks perform well on small datasets if we spend enough time on hyper-parameter tuning. A comparison of the ImageNet dataset is more sensible but requires intensive resources for hyper-parameter tuning. The common practice is assuming the related works already report the best results they can produce, comparing our ImageNet results with related works, and showing that our result is reproducible with a standard training pipeline. We released our training code based on the official PyTorch ImageNet example, and Table 1&2 show the superiority of our method compared to other ternary/shift networks. The ablation study shows that our method consistently outperforms other shift network training methods under a wide range of hyper-parameters.
>
> Secondly, **Weight Sign Variation Rate (WSVR) is not a score.** As mentioned by reviewer FA7K and CUjS, Weight Sign Variation Rate is an indicator used to depict the learning dynamics of weight. Intuitively, it is an indicator of easy it is to flip the weight sign by the gradient update signal. A large value of WSVR only makes sense while following gradient update during network training while minimizing a certain loss simultaneously.
>
> To recap our point:
> 1)	High WSVR during training indicates the weight sign can be easily flipped by the gradient update signal when needed, and we observe a high WSVR during the full-precision network training. We suppose this property is beneficial to the convergence speed of the optimizer.
> 2)	We observed the ternary networks trained with a quantizer appear with a low WSVR. Two potential answers could be: (a) weight sign flipping is unnecessary for ternary weight learning (b) weight sign cannot quickly flip in the optimization space defined by the traditional ternary quantizer.
> 3)	Fig 2 weight histogram visualization implied the ternary quantizer prevents the weight sign variation during training (discussed in session 3.2). Therefore, we design S3 re-parameterization that seem effective, and the ternary weight can flip their sign freely in this new optimization space. We confess that this phenomenon requires more theoretical understanding and we plan to follow up with a theory paper in the future.
> 4)	We observed the ternary weight sign is flipping rapidly (high WSVR) during training with S3 re-parameterization. This observation supports the weight sign flipping is necessary for ternary weight learning, and the answer (b) is correct, (a) is simply wrong.
>
> #### **Q5:** The vanishing gradient problem described in section 3.1 is only valid when eq (2) is a scalar multiplication chain. However, in a matrix multiplication chain, the final product is not necessarily zero/sparse. Requires more evidence to support this is a real issue.
>
> #### **A5:**  In the camera-ready version, we will elaborate more on the vanishing gradient problem (VGP) description in section 3.1. Here we clarify our VGP definition in its matrix multiplication form.
> **The VGP discussed in section 3.1 is the backprop gradient information loss due to one or more terms equal to zero in the backward propagation multiplication chain.** We are the first to point out that the sparse weight in a low-bit quantized network will lead to this VGP and propose s3 reparameterization as a remedy. The sparse derivative of ReLU leads to a very similar VGP in the ReLU-activated network, which is widely known as dying ReLU. As mentioned in several related works [4,5], dying ReLU is regarded as a special vanishing gradient problem.
>
> Let's use a simple example to clarify the VGP definition in this paper further. Consider a 2x2 matrix multiplication with ReLU activation from layer l to l+1,
>
> $\begin{pmatrix} x_{0}^{l+1}, \\\\  x_{1}^{l+1} \end{pmatrix} =  \begin{pmatrix}
>         \text{ReLU}(W_{00}^{l}x_{0}^{l} + W_{01}^{l}x_{1}^{l}),  \\\\
>         \text{ReLU}(W_{10}^{l}x_{0}^{l} + W_{11}^{l}x_{1}^{l})
>         \end{pmatrix}$
>
> The backward gradient update toward $x_{0}^{l}$ is computed as below,
>
> $\frac{\partial \text{Loss}}{\partial x_{0}^{l}}  = \frac{\partial \text{Loss}}{\partial x_{0}^{l+1}} \frac{\partial x_{0}^{l+1}}{\partial (W_{00}^{l}x_{0}^{l} + W_{01}^{l}x_{1}^{l})} W_{00}^{l} + \frac{\partial \text{Loss}}{\partial x_{1}^{l+1}}\frac{\partial x_{1}^{l+1}}{\partial (W_{10}^{l}x_{0}^{l} + W_{11}^{l}x_{1}^{l})} W_{10}^{l} $
>
> Where $ \frac{\partial x_{1}^{l+1}}{\partial (W_{10}^{l}x_{0}^{l} + W_{11}^{l}x_{1}^{l})}$ is the derivative of ReLU.
> $\frac{\\partial x_{1}^{l+1}}{\partial (W_{10}^{l}x_{0}^{l} + W_{11}^{l}x_{1}^{l})}=\left \\{ \begin{aligned}1 & & W_{10}^{l}x_{0}^{l} + W_{11}^{l}x_{1}^{l} > 0 \\\\ 0 & & W_{10}^{l}x_{0}^{l} + W_{11}^{l}x_{1}^{l} \leq 0 \end{aligned}\right. $
>
>
> The equations above show that the gradient information $\frac{\partial \text{Loss}}{\partial x_{1}^{l+1}}$ propagating from $x_{1}^{l+1}$ to $x_{0}^{l}$ vanishes in two situations. (1) the corresponding derivative of ReLU is zero. (i.e. the neuron is not activated) (2) the corresponding weight value $W_{10}^{l}$ is zero.
>
> Dying ReLU issue refers to when (1) happens on all training examples. In this situation, the gradient will vanish forever, and a specific portion of weight will never receive gradient updates regardless of the feeding training examples. Up to 40\% of the network can be 'die' in ReLU-activated networks, and these weights cannot be sufficiently adjusted by the backprop gradient update signal [6] And (2) is the VGP discussed in section 3.1 of this work. We argue that (2) is much easier to occurs than (1) since the gradient vanishes once the corresponding weight value is zero, regardless of the input training example.
>
>
> Reference:
>
> [1] Fengfu Li, Bo Zhang, and Bin Liu. Ternary weight networks.arXiv preprint arXiv:1605.04711, 2016.
>
> [2] Xiang Deng and Zhongfei Zhang. An embarrassingly simple approach to training ternary weight networks.arXiv preprint arXiv:2011.00580, 2020.
>
> [3] Chenzhuo Zhu, Song Han, Huizi Mao, and William J Dally. Trained ternary quantization.arXivpreprint arXiv:1612.01064, 2016.
>
> [4] Zheng Hu, Jiaojiao Zhang, and Yun Ge.  Handling vanishing gradient problem using artificial derivative.IEEE Access, 9:22371–22377, 2021.
>
> [5]Lu Lu, Yeonjong Shin, Yanhui Su, and George Em Karniadakis. Dying relu and initialization: Theory and numerical examples.arXiv preprint arXiv:1903.06733, 2019.
>
> [6] CS231N course note: https://cs231n.github.io/neural-networks-1/

---

> > ### Author Response · Authors · 2021-08-11
> > **Additional response**
> >
> >
> > #### **Q6:** The discrete weight optimized by a quantizer will oscillate around the thresholding points. In the case of ternary quantizer, the weights are oscillating between -1, 0 and 0, 1. However, for higher bit quantizer (for instance, -3,-2,-1,0,1,2,3), the weights will oscillate around other thresholding points such as between 2, 3. The sign is not specific to this problem.
> >
> > #### **A6:** We agree that a similar issue might also affect higher bit quantizers, which requires further investigation. We plan to generalize the study of weight sign freezing to higher bit quantizer in the follow-up research. However, we focus on improving the ternary quantizer in this paper since it is a key design in many training solutions of low-bit weight networks, such as TWN [1], TTQ [3], DeepShift [7] and INQ[8].
> >
> >
> > [7] Mostafa Elhoushi, Zihao Chen, Farhan Shafiq, Ye Henry Tian, and Joey Yiwei Li. Deepshift: Towards multiplication-less neural networks.arXiv preprint arXiv:1905.13298, 2019.
> >
> > [8] Aojun Zhou, Anbang Yao, Yiwen Guo, Lin Xu, and Yurong Chen. Incremental network quantization: Towards lossless cnns with low-precision weights.arXiv preprint arXiv:1702.03044,1892017.

---

> > ### Comment · Reviewer_KeAd · 2021-08-26
> > **Further discussion**
> >
> > Dear Authors,
> >
> > Thank you for the response.
> >
> > I want to hone in on a very specific thing of the method with you, the reparametrization of the ternary function.
> >
> > Let's say I draw a w_sparse, w_sign diagram (x, y).
> > That state-space would, when applying the heaviside functions with some very poor ascii drawing look like this:
> >
> > 0  |  1
> > ——
> > 0  | -1
> >
> > (edit: ok, ascii drawing doesn't work... imagine a plot with 4 quadrants, 0 in the top left, 1 in the top right, 0 on bottom left, -1 bottom right.)
> >
> > Now I want to see what is happening when I get a certain gradient that my w_ter(w_sparse, w_sign) should be updated into (higher or lower), and express what that gradient update would do to w_sparse and w_sign. We either walk up, or down in the state space based on the gradients.
> > We can interpret this as a linear ODE, as follows:
> > dx\dt = 2*H(y)-1
> > dy\dt = 2H(x)
> >
> > with x and y as above. The solutions of these can be visualized with this tool: https://www.wolframalpha.com/widgets/view.jsp?id=9298fea31cf266903b3df7174b95ddd7. (or any other phase-space tool that visualizes solutions to ODEs!)
> > If you fill in the equations above, and let's say a range of -5 to 5, you'll see solution lines.
> >
> > The solution lines in blue, indicate the trajectories that your solutions would follow (if the gradients and the updates were infinitely small) theoretically. These depend on the initial condition mostly. These line solutions are logical, since you are parametrizing 1 float value with 2 float values, the differences in the 1 float value w_ter have to follow a 1-dimensional subspace of the space of the two float values. The solutions never cross the y=-x and y=x nullclines.
> >
> > Based on this phase diagram, assuming small 0-mean initialization, and barring inaccuracies because we're setting non-infinitesimal steps, it looks like your method is mostly doing the following, depending on the region between the nullclines they are defined between:
> > 1/3rd of your values move between the ternary values of 0 and 1,
> > another 1/3rd change between -1 and 1,
> > another 1/3rd change between -1 and 0,
> >
> > It would be great if you could show me either where my analysis is wrong, or show some 'traces' of weights and how they update over time and how they switch.
> > If I'm correct, your method could just as well be a method where some weights switch between 0-1, some -1-1 some -1-0.
> >
> > Adding the weight densification term, if the alpha is large enough this pulls weights towards the (-1, 1) region, meaning they will never be 0 again. I wonder if you couldn't get the same results if you had no 0s, but only optimized binary values between -1 and 1 in your setup.

---

> > > ### Author Response · Authors · 2021-08-30
> > > **Re: Further discussion**
> > >
> > > Thank you for the new comment! We summarized your question below, and here is our answer.
> > >
> > > **Q:** The variation trace of w_sparse and w_sign can be interpret as an linear ODE (dx\dt = 2*(y)-1 dy\dt = 2x). When d_loss/dW_ter is positive (want to go up), w_sparse and w_sign move along the blue curves toward the arrow direction. When d_loss/dW_ter is negative (want to go down), w_sparse and w_sign move along the blue curves toward the opposite arrow direction. The visual interpretation suggests that the w_sparse and w_sign will only osccilating between three value pairs {-1,1}, {0,1} or {0,-1} depends on their initialization values.
> > >
> > > **A:** We read the analysis carefully and find that your conclusion is wrong because of the wrong assumption, and it does not match our experimental observation.
> > >
> > > Firstly, we think the linear ODE you provide is inaccurate. The heaviside functions are missing in the linear ODE.
> > >
> > > The correct formulas are
> > >
> > > dx\dt = 2*UnitStep(y)-1,    dy\dt = UnitStep(x)
> > >
> > > Secondly, **as you mentioned, w_sparse and w_sign follow the solution lines in blue if their gradients and updates were infinitely small,  and assume the inaccuracies of non-infinitesimal steps can ignore. However, this is a wrong assumption during training.**
> > >
> > > The weight update is the product of (1) the local negative gradient and (2) the current learning rate. However, none of these terms should be assumed infinitesimals during training.  The local gradient is noisy due to the stochastic gradient method is used. The learning rate is not infinitesimal. Otherwise, the network can not learns. **w_sparse and w_sign follows the solution lines only when the learning rate is infinitesimal, so we should not ignore the inaccuracies of non-infinitesimal steps in this analysis**
> > >
> > > Thirdly, **if the learning rate is not infinitesimal, we must consider the inaccuracies of the non-infinitesimal steps. In this case, the ternary weights can switch between -1,0 and 1.**
> > >
> > > The inaccuracies are similar to the error of solving the ODE with Euler's method. In this case, w_sparse and w_sign over-shoot along the tangent line of blue curves.  Here, we provide a figure to demonstrate the weight traces when the learning rate is non-infinitesimal.
> > >
> > > https://1drv.ms/u/s!AvGUsxgZ9_H5gtAL309dyTbCg2sEww?e=k3dlAR
> > >
> > > The "non-infinitesimal steps' inaccuracies" cause an over-shooting effect on weight movement trace. As shown in our figure, the over-shooting effect pushes any weights oscillating between {-1,1}, {0,1} and {0,-1} toward the origin point of the decomposed optimization space (see the arrows in dark red). With the over-shooting effect, the weights can easily switch between -1, 0 and 1 when they stay close to the origin point. (see the arrows in red)
> > >
> > > **In conclusion, your analysis is valid when the learning rate is infinitesimal. When the learning rate is non-infinitesimal, the ternary weights can switch between -1, 0 and 1 regardless of their initial values**

---

> > > > ### Author Response · Authors · 2021-09-17
> > > > **Re2: Further discussion**
> > > >
> > > > Dear reviewer KeAd,
> > > >
> > > > To further clarify your concern, we have prepared further analysis on the trajectory of the weights. **The trajectory of the ternary weights in the decomposition space (w_sign and w_sparse) during the actual training clearly shows that the conclusion of your theoretical analysis is incorrect.**
> > > >
> > > > **We first explain and justify the setup of creating these visualizations.** We ran the ternary CIFAR10 experiment described in section 5.2 again and saved the weight snapshots at the beginning of each epoch. We randomly sampled multipe 3x3 filters in ResNet-20 and plotted the trajectories of their nine weight elements during training.
> > > >
> > > > The trajectories are similar so we choose one of the 3x3 filter to visualize. The reasons for choosing this particular filter are as follows:
> > > > 1. The nine elements are initialized in four different quadrants, which are scattered enough to show the trajectories in each quadrant.
> > > > 2. One of the elements flips the signs of w_sign and w_sparse multiple times during the training process. We want to show the weight trajectory under frequent sign flipping.
> > > >
> > > > Figure 1 shows the trajectories of the nine elements during training. In the figure, the nine elements represent by different colours. In the decomposition space, the three regions separated by w_sparse=w_sign, w_sparse=-w_sign and w_sign=0 are also filled with different colours to analyze the trajectory better. **If your conclusion is correct, the weight trajectory can never move from one region to another.** Furthermore, every time the trajectory crosses the boundary between the regions, we mark the current epoch number next to the trajectory.
> > > >
> > > > Here is [Figure 1: The weight trajectories of a typical 3x3 filter during training (w_sign, w_sparse=-0.075~0.075)](https://1drv.ms/u/s!AvGUsxgZ9_H5gtANnsYjpOofOHz9Zw?e=MSDvUc)
> > > >
> > > > It can be seen from figure 1 that regardless of the initial value, with training, the ternary weights move to the origin of the decomposition space.
> > > >
> > > > Then obtain figure 2 by further zoom in the area close to the origin.
> > > >
> > > > Here is [Figure 2: The weight traces of a typical 3x3 filter during training (w_sign, w_sparse=-0.001~0.001)](https://1drv.ms/u/s!AvGUsxgZ9_H5gtAOjLP5-26rd1a90g?e=mkuyxm)
> > > >
> > > > Visualize only one of the elements in the 3x3 filter to obtain Figure 3
> > > >
> > > > Here is [Figure 3: The trace of the quickly flipping weight during training (w_sign, w_sparse=-0.001~0.001)](https://1drv.ms/u/s!AvGUsxgZ9_H5gtAMsff08n1S-ggVAg?e=v0JBYz)
> > > >
> > > > **It can be seen from Figures 2 and 3 that when the ternary weight is close to the origin of the decomposition space, the trajectory can easily cross the boundary between the three regions. This is contrary to the conclusion of your analysis that the weight trajectory cannot cross these boundaries.**
> > > >
> > > > Figures 1, 2, and 3 further support our previous theoretical analysis conclusion, summarized in the [previous figure](https://1drv.ms/u/s!AvGUsxgZ9_H5gtAL309dyTbCg2sEww?e=k3dlAR).
> > > >
> > > > Secondly, **your analysis conclusion contradicts the pre-trained checkpoints we released earlier in our author response to reviewer geT6.** If your conclusion is correct, then after 200 epochs training, there should be no 0 value in the weight. However, in our pre-training models, the weights of each layer still contain zero values.
> > > >
> > > > Here we provide a script to quickly verify that each layer's weight in the pre-training checkpoint contains zero values.
> > > >
> > > > https://1drv.ms/u/s!AvGUsxgZ9_H5gtAPBwijdX6YMQNdLw?e=MAvtOu

---

> ### Author Response · Authors · 2021-08-25
> **Additional result and new table**
>
> We update Table 1 with 4-bit result and add more recent related works for a better comparison. **S3-ShiftNN (4bit) reaches 70.47% Top1 accuracy on ImageNet.**
>
> |       Kernel      |    Methods    | Bit-width | Initialization | Top1 Accu. (%) | Top5 Accu. (%) |
> |:-----------------:|:-------------:|:---------:|:--------------:|:--------------:|:--------------:|
> |       Multi.      |      FP32     |     32    |     Random     |      69.6      |      89.2      |
> |       Multi.      |     TTQ[4]    |     2     |   Pre-trained  |      66.6      |      87.2      |
> | Sum of Sign Flips |   Lq-Net[1]   |     2     |     Random     |      68.0      |      88.0      |
> | Sum of Sign Flips |   Lq-Net[1]   |     3     |     Random     |      69.3      |      88.8      |
> | Sum of Sign Flips |   Lq-Net[1]   |     4     |     Random     |      70.0      |      89.1      |
> | Sum of Xnor-Popcount |   ABC-Net[2]  |     5     |     Random     |      68.3      |      87.9      |
> |     Sign Flip     |     BWN[5]    |     1     |     Random     |      60.8      |      83.0      |
> |     Sign Flip     |    HWGQ[7]    |     1     |     Random     |      61.3      |      83.2      |
> |     Sign Flip     |    BWHN[8]    |     1     |     Random     |      64.3      |      85.9      |
> |     Sign Flip     |   IR-net[10]   |     1     |     Random     |      66.5      |      86.8      |
> |     Sign Flip     |     TWN[6]    |     2     |     Random     |      61.8      |      84.2      |
> |     Sign Flip     |   LR-net[3]   |     2     |     Random     |      63.5      |      84.8      |
> |     Sign Flip     |   SQ-TWN[9]  |     2     |     Random     |      63.8      |      85.7      |
> |     Sign Flip     |    INQ[12]    |     2     |   Pre-trained  |      66.02     |      87.13     |
> |     Sign Flip     |      Ours     |     2     |     Random     |      66.37     |      87.18     |
> | Shift + Sign Flip |    INQ[12]    |     3     |   Pre-trained  |      68.08     |      88.36     |
> | Shift + Sign Flip |    INQ[12]    |     4     |   Pre-trained  |      68.89     |      89.01     |
> | Shift + Sign Flip |    INQ[12]    |     5     |   Pre-trained  |      68.98     |      89.10     |
> | Shift + Sign Flip | DeepShift[11] |     6     |     Random     |      65.63     |      86.33     |
> | Shift + Sign Flip | DeepShift[11] |     6     |   Pre-trained  |      68.32     |      88.41     |
> | Shift + Sign Flip |      Ours     |     3     |     Random     |      69.59     |      89.22     |
> | Shift + Sign Flip |      Ours     |     4     |     Random     |      70.47     |      89.93     |
>
> 'Sum of Sign Flips' methods achieved similar accuracies to our method. However, they are more expensive to compute. For an n-bit weight, this kernel requires n sign-flip operations and (n-1) addition operations to compute the sum of sign-flips. Our method only requires one sign flip operation and one shift operation for n-bit weight. Note that the energy cost of addition operation and shift operation are similar on ASIC and FPGA [13].
>
> IR-net is a powerful method to train a 1bit network. The two optimization difficulties discussed in this paper: weight sign freezing and gradient vanishing, do not apply to a 1bit network. However, like many other 1bit networks, IR-net suffers from the accuracy drop caused by its low-bit, and it can not scale to a higher bit for better accuracy. When accuracy is necessary, we have to modify the network architecture to reduce the accuracy gap compare to the full-precision model. A commonly-used solution is network widening[14], which is expensive. A 2x widening requires 2x activation memory occupation, 4x weight memory occupation, 4x storage size and 4x operations. It can quickly mitigate the performance benefit of 1-bit methods.
>
> [1] Dongqing Zhang, Jiaolong Yang, Dongqiangzi Ye, and Gang Hua. Lq-nets: Learned quantization for highly accurate and compact deep neural networks. In Proceedings of the European conference on computer vision (ECCV), pages 365–382, 2018.
>
> [2] Xiaofan Lin, Cong Zhao, and Wei Pan. Towards accurate binary convolutional neural network.arXiv preprint arXiv:1711.11294, 2017.
>
> [3] Oran Shayer, Dan Levi, and Ethan Fetaya. Learning discrete weights using the local reparameterization trick. arXiv preprint arXiv:1710.07739, 2017.
>
> [4] Chenzhuo Zhu, Song Han, Huizi Mao, and William J Dally. Trained ternary quantization.arXiv preprint arXiv:1612.01064, 2016.
>
> [5] Mohammad Rastegari, Vicente Ordonez, Joseph Redmon, and Ali Farhadi. Xnor-net: Imagenet classification using binary convolutional neural networks. InEuropean conference on computer vision, pages 525–542. Springer, 2016.
>
> [6] Fengfu Li, Bo Zhang, and Bin Liu. Ternary weight networks.arXiv preprint arXiv:1605.04711, 2016.
>
> [7] Zhaowei Cai, Xiaodong He, Jian Sun, and Nuno Vasconcelos. Deep learning with low precision by half-wave gaussian quantization. InProceedings of the IEEE conference on computer vision and pattern recognition, pages 5918–5926, 2017.
>
> [8] Qinghao Hu, Peisong Wang, and Jian Cheng. From hashing to cnns: Training binary weight networks via hashing. In Thirty-Second AAAI conference on artificial intelligence, 2018.
>
> [9] Yinpeng Dong, Renkun Ni, Jianguo Li, Yurong Chen, Jun Zhu, and Hang Su. Learning accurate low-bit deep neural networks with stochastic quantization.arXiv preprint arXiv:1708.01001, 2017.
>
> [10] Haotong Qin, Ruihao Gong, Xianglong Liu, Mingzhu Shen, Ziran Wei, Fengwei Yu, and Jingkuan Song. Forward and backward information retention for accurate binary neural networks. In CVPR, 2020.
>
> [11] Mostafa Elhoushi, Zihao Chen, Farhan Shafiq, Ye Henry Tian, and Joey Yiwei Li. Deepshift: Towards multiplication-less neural networks.arXiv preprint arXiv:1905.13298, 2019.
>
> [12] Aojun Zhou, Anbang Yao, Yiwen Guo, Lin Xu, and Yurong Chen. Incremental network quantization: Towards lossless cnns with low-precision weights.arXiv preprint arXiv:1702.03044,1892017.
>
> [13] You, Haoran, et al. "Shiftaddnet: A hardware-inspired deep network." arXiv preprint arXiv:2010.12785 (2020).
>
> [14] McDonnell MD. Training wide residual networks for deployment using a single bit for each weight. arXiv preprint arXiv:1802.08530. 2018 Feb 23.

---

### Official Review · Reviewer_geT6 · 2021-07-04

**Rating:** 8
**Confidence:** 3

**Summary:**

The paper suggests a new training technique for neural networks, where each parameter is broken down into multiple parameters, representing its sign, sparsity, and magnitude (or shift). They explain why this new technique is useful for overcoming vanishing gradients and sign-freezing. They show that this training can produce a network with 3 bits weights, with the same quality as the standard fp32 network.

**Limitations And Societal Impact:**

1. My main concern with the paper is reproducibility. Without additional details and/or code, the experimental part can not be properly evaluated.

2.  Please highlight that the benefit to network efficiency is for inference only, and specify the cost for the technique during training. The limitations should be clearly mentioned in either the introduction or the conclusion.

3. For the ablation study to be more effective, it should scan the same area as the final result- please increase the number of epochs to 200. It will also be helpful to measure the standard deviation of the main results.

4. Regardless of reproducibility, section 4 should be expanded. Since reparameterization of every value utilizes Heaviside functions, the values of $w_{sign}, w_{sparse}, w_{n}$ can not be optimized with standard backpropagation. According to Figure 3, STEs are used, but this is not explained in the text as far as I can tell. The issue of initializing these parameters is also not mentioned-- initialization might have important consequences since sign freeze and (sparse-freeze) can still occur if the values were not initialized properly.

(Minor)

5. The data in Table 4 would be clearer if visualized in a plot with the familiar format (Top1 accuracy vs Epochs). If space is a concern, I think it would be preferable to have a figure visualizing it in the appendix than a table within the main text.

6. The front page claim of out-performing full-precision networks is exaggerated. For it to be made, I would expect clear error bars for both results, proving that the result is significant. The ablation study implies the standard deviation is on par with the current accuracy gap.

7. Expanding table 1 with more baselines will give the reader a better picture of the tradeoffs between different approaches.

**Main Review:**

**Originality**:

The method suggested in the paper is novel. Related work is adequately cited.

**Quality**

The submission is technically sound.

The paper starts by discussing the problems of training low-bit networks. The paper claims sign-freezing and vanishing gradients are at fault, and provide some evidence that this is the case.

The next part discusses the proposed technique, of S3 reparameterization. I found the details of the implementation of sign-sparse-shift training to be lacking. (See limitation 3,4)

The experimental results in the paper cover the necessary benchmarks, showing benefits for both resnet18 and resnet50 with Imagenet. The baselines give a relatively good representation of competing methods. I would suggest adding several more baselines for a clearer picture: I was using this survey [1] for comparison, and some of the lines seem to also be relevant for this paper as well.

While the competition on reducing bit width is stiff (And may also include activation width, where this approach may be less appealing), I consider the ability to train an efficient network from scratch to be this paper's great advantage, as most papers that used low bits implementations rely heavily on a pre-trained model.

The ablation study was very helpful in showing the robustness of the new technique. However, all the results in the study were lower than the published result by ~0.5%, including the marked result ($\alpha = 1e-5$, No scheduler). This can be explained by the reduced number of epochs, but its means that the ablation study covers an area that is somewhat different from the one we are interested in.

Of course, the paper can still benefit from the inclusion of more benchmarks to show that the results generalize well to other domains. The sensitivity analysis is useful at verifying that the S3 training indeed solves the problem it was designed to.

One big limitation of the paper is the reparameterization of the weights during training is expected to be costly (in memory, at least). I don't think this limitation has been properly addressed, and I would have missed it in my initial read if it wasn't noted in lines 199-200.

**Clarity:**

Overall, the paper is clearly written.
As mentioned above, some details that can help a reader reproduce the work are missing.

**Significance**
The results are very important, and pending further research, have the potential to be translated to more efficient hardware for inference. As a successful training technique for low bits width, this work may also inspire theoretical works that can further advance our understanding of the theory of learning.

[1] https://arxiv.org/abs/2101.09671

**Time Spent Reviewing:**

5

---

> ### Author Response · Authors · 2021-08-10
> **Author response**
>
> Thank you for taking the time to review our paper and summarizing the main idea. Thank you for describing our paper as well-written, technically sound and recognizing novelty and impressive results as the main strengths of this work.
>
> It appears you capture the shining point of this work that our method significantly reduces the weight bit-width (5bit -> 3bit) while improving the ImageNet classification accuracy of shift neural networks at the same time. Also, thank you for recognizing the paper would be a contribution to the literature.
>
> $\def\wshift{w_\mathrm{shift}}
> \def\wsparse{w_\mathrm{sparse}}
> \def\wsign{w_\mathrm{sign}}
> \def\wter{w_\mathrm{ter}}$
>
> However, you express valid concerns we would like to address. We believe all points can be addressed in a revision to the paper.
>
> #### **Q1:** The main concern is reproducibility. Without additional details and/or code, the experimental part can not be properly evaluated.
>
> #### **A1:** To address your concern about the reproducibility of this paper, we release our training code and pre-trained checkpoint for further verification. The code is not released with the original submission since it was an intellectual property legal process ongoing and we can not get the code release approval before the paper submission deadline.
>
> Click [here](https://1drv.ms/u/s!AvGUsxgZ9_H5gs9_vLhjPmJEXZ2yNQ?e=bCDENZ) to download.
>
> #### **Q2:** The Table 3 ablation study was helpful, but it is not trained with full epochs (120 epochs vs 200 epochs).
>
> #### **A2:** We run the experiments in Table 3 again with 200 epochs, and here are the new experimental results. We will use this new table in the camera-ready version.
>
> | Decay Scheduler | ResNet18 ImageNet | $\alpha$=1e-2 | 1e-3 | 1e-4 | 1e-5 | 1e-6 |
> |---|---|---|---|---|---|---|
> | None | Top1 / Top5 | 69.78 / 89.21 | 69.85 / 89.24 | 69.84 / 89.23 | 69.82 / 89.23 | 69.83 / 89.23 |
> | Linear | Top1 / Top5 | 69.91 / 89.23 | 69.68 / 89.06 | 69.74 / 89.13 | 69.62 / 89.10 | 66.70 / 86.93 |
> | Cosine | Top1 / Top5 | 69.91 / 89.15 | 69.85 / 89.20 | 69.85 / 89.38 | 69.57 / 89.04 | 65.82 / 86.50 |
>
> #### **Q3:** One big limitation of this paper is the training solution is costly (at least in memory). And the benefit of this method is for inference only.
>
> #### **A3:** We will highlight the limitation of this method in the conclusion of the camera-ready version. Although the S3 re-parameterization increases memory occupation during training compared to the traditional quantizer, we would argue that it is not as costly as its first impression, especially for CNN training. Sohoni et al. [1] show that the activation memory dominates memory occupation during ResNet and Transformer architecture training. Figure 1 shows the combined model and optimizer memory only occupy 4.5\% of total memory consumption of ResNet and 21.3\% of total memory consumption of Transformer during training [1]. The extra computational cost and memory occupation of S3 re-parameterization depends on the size of the weight, not the activation.
>
> #### **Q4:** More baselines should be added to Table 1\&2.
>
> #### **A4:** More baselines [2, 3, 4, 5, 6, 7] are included in our first draft. They are removed from the final submission due to page limitations. We will re-allocate part of the ablation study to the appendix to save some space and bring these experimental results back to Tables 1/2 in the camera-ready version. Important missing baselines will be added in the camera-ready version.
>
>
> #### **Q5:** More implementation details should add to session 4.
>
> #### **A5:** We will add more implementation details to session 4, including initialization strategy. And the implementation details can be verified in our code.
>
>
> #### **Q6:** The front page claim of out-performing full-precision networks is exaggerated.
>
> #### **A6:** We will replace 'out-performing' with 'competing' in the camera-ready version to address your concern.
>
> Reference:
>
> [1] Nimit  Sharad  Sohoni,  Christopher  Richard  Aberger,  Megan  Leszczynski,  Jian  Zhang,  and Christopher R’e.  Low-memory neural network training:  A technical report.arXiv preprint arXiv:1904.10631, 2019.
>
> [2] Dongqing Zhang, Jiaolong Yang, Dongqiangzi Ye, and Gang Hua. Lq-nets: Learned quantization for highly accurate and compact deep neural networks. In Proceedings of the European conference on computer vision (ECCV), pages 365–382, 2018.
>
> [3] Yuhang Li, Xin Dong, and Wei Wang.  Additive powers-of-two quantization: An efficient non-uniform discretization for neural networks.arXiv preprint arXiv:1909.13144, 2019.
>
> [4] Xiaofan Lin, Cong Zhao, and Wei Pan. Towards accurate binary convolutional neural network.arXiv preprint arXiv:1711.11294, 2017.
>
> [5] Matthieu Courbariaux, Yoshua Bengio, and Jean-Pierre David. Binaryconnect: Training deep neural networks with binary weights during propagations.arXiv preprint arXiv:1511.00363, 2015.
>
> [6] Zhouhan Lin, Matthieu Courbariaux, Roland Memisevic, and Yoshua Bengio. Neural networks with few multiplications. arXiv preprint arXiv:1510.03009, 2015.
>
> [7] Oran Shayer, Dan Levi, and Ethan Fetaya. Learning discrete weights using the local reparameterization trick. arXiv preprint arXiv:1710.07739, 2017.

---

> > ### Author Response · Authors · 2021-08-25
> > **New table and 4bit result**
> >
> > **A4:** We update Table 1 with 4-bit result and add more recent related works for a better comparison. **S3-ShiftNN (4bit) reaches 70.47% Top1 accuracy on ImageNet.**
> >
> > |       Kernel      |    Methods    | Bit-width | Initialization | Top1 Accu. (%) | Top5 Accu. (%) |
> > |:-----------------:|:-------------:|:---------:|:--------------:|:--------------:|:--------------:|
> > |       Multi.      |      FP32     |     32    |     Random     |      69.6      |      89.2      |
> > |       Multi.      |     TTQ[4]    |     2     |   Pre-trained  |      66.6      |      87.2      |
> > | Sum of Sign Flips |   Lq-Net[1]   |     2     |     Random     |      68.0      |      88.0      |
> > | Sum of Sign Flips |   Lq-Net[1]   |     3     |     Random     |      69.3      |      88.8      |
> > | Sum of Sign Flips |   Lq-Net[1]   |     4     |     Random     |      70.0      |      89.1      |
> > | Sum of Xnor-Popcount |   ABC-Net[2]  |     5     |     Random     |      68.3      |      87.9      |
> > |     Sign Flip     |     BWN[5]    |     1     |     Random     |      60.8      |      83.0      |
> > |     Sign Flip     |    HWGQ[7]    |     1     |     Random     |      61.3      |      83.2      |
> > |     Sign Flip     |    BWHN[8]    |     1     |     Random     |      64.3      |      85.9      |
> > |     Sign Flip     |   IR-net[10]   |     1     |     Random     |      66.5      |      86.8      |
> > |     Sign Flip     |     TWN[6]    |     2     |     Random     |      61.8      |      84.2      |
> > |     Sign Flip     |   LR-net[3]   |     2     |     Random     |      63.5      |      84.8      |
> > |     Sign Flip     |   SQ-TWN[9]  |     2     |     Random     |      63.8      |      85.7      |
> > |     Sign Flip     |    INQ[12]    |     2     |   Pre-trained  |      66.02     |      87.13     |
> > |     Sign Flip     |      Ours     |     2     |     Random     |      66.37     |      87.18     |
> > | Shift + Sign Flip |    INQ[12]    |     3     |   Pre-trained  |      68.08     |      88.36     |
> > | Shift + Sign Flip |    INQ[12]    |     4     |   Pre-trained  |      68.89     |      89.01     |
> > | Shift + Sign Flip |    INQ[12]    |     5     |   Pre-trained  |      68.98     |      89.10     |
> > | Shift + Sign Flip | DeepShift[11] |     6     |     Random     |      65.63     |      86.33     |
> > | Shift + Sign Flip | DeepShift[11] |     6     |   Pre-trained  |      68.32     |      88.41     |
> > | Shift + Sign Flip |      Ours     |     3     |     Random     |      69.59     |      89.22     |
> > | Shift + Sign Flip |      Ours     |     4     |     Random     |      70.47     |      89.93     |
> >
> > 'Sum of Sign Flips' methods achieved similar accuracies to our method. However, they are more expensive to compute. For an n-bit weight, this kernel requires n sign-flip operations and (n-1) addition operations to compute the sum of sign-flips. Our method only requires one sign flip operation and one shift operation for n-bit weight. Note that the energy cost of addition operation and shift operation are similar on ASIC and FPGA [13].
> >
> > IR-net is a powerful method to train a 1bit network. The two optimization difficulties discussed in this paper: weight sign freezing and gradient vanishing, do not apply to a 1bit network. However, like many other 1bit networks, IR-net suffers from the accuracy drop caused by its low-bit, and it can not scale to a higher bit for better accuracy. When accuracy is necessary, we have to modify the network architecture to reduce the accuracy gap compare to the full-precision model. A commonly-used solution is network widening[14], which is expensive.  A 2x widening requires 2x activation memory occupation, 4x weight memory occupation, 4x storage size and 4x operations. It can quickly mitigate the performance benefit of 1-bit methods.
> >
> > [1] Dongqing Zhang, Jiaolong Yang, Dongqiangzi Ye, and Gang Hua. Lq-nets: Learned quantization for highly accurate and compact deep neural networks. In Proceedings of the European conference on computer vision (ECCV), pages 365–382, 2018.
> >
> > [2] Xiaofan Lin, Cong Zhao, and Wei Pan. Towards accurate binary convolutional neural network.arXiv preprint arXiv:1711.11294, 2017.
> >
> > [3] Oran Shayer, Dan Levi, and Ethan Fetaya. Learning discrete weights using the local reparameterization trick. arXiv preprint arXiv:1710.07739, 2017.
> >
> > [4] Chenzhuo Zhu, Song Han, Huizi Mao, and William J Dally. Trained ternary quantization.arXiv preprint arXiv:1612.01064, 2016.
> >
> > [5] Mohammad Rastegari, Vicente Ordonez, Joseph Redmon, and Ali Farhadi. Xnor-net: Imagenet classification using binary convolutional neural networks. InEuropean conference on computer vision, pages 525–542. Springer, 2016.
> >
> > [6] Fengfu Li, Bo Zhang, and Bin Liu. Ternary weight networks.arXiv preprint arXiv:1605.04711, 2016.
> >
> > [7] Zhaowei Cai, Xiaodong He, Jian Sun, and Nuno Vasconcelos. Deep learning with low precision by half-wave gaussian quantization. InProceedings of the IEEE conference on computer vision and pattern recognition, pages 5918–5926, 2017.
> >
> > [8] Qinghao Hu, Peisong Wang, and Jian Cheng. From hashing to cnns: Training binary weight networks via hashing. In Thirty-Second AAAI conference on artificial intelligence, 2018.
> >
> > [9] Yinpeng Dong, Renkun Ni, Jianguo Li, Yurong Chen, Jun Zhu, and Hang Su. Learning accurate low-bit deep neural networks with stochastic quantization.arXiv preprint arXiv:1708.01001, 2017.
> >
> > [10] Haotong Qin, Ruihao Gong, Xianglong Liu, Mingzhu Shen, Ziran Wei, Fengwei Yu, and Jingkuan Song. Forward and backward information retention for accurate binary neural networks. In CVPR, 2020.
> >
> > [11] Mostafa Elhoushi, Zihao Chen, Farhan Shafiq, Ye Henry Tian, and Joey Yiwei Li. Deepshift: Towards multiplication-less neural networks.arXiv preprint arXiv:1905.13298, 2019.
> >
> > [12] Aojun Zhou, Anbang Yao, Yiwen Guo, Lin Xu, and Yurong Chen. Incremental network quantization: Towards lossless cnns with low-precision weights.arXiv preprint arXiv:1702.03044,1892017.
> >
> > [13] You, Haoran, et al. "Shiftaddnet: A hardware-inspired deep network." arXiv preprint arXiv:2010.12785 (2020).
> >
> > [14] McDonnell MD. Training wide residual networks for deployment using a single bit for each weight. arXiv preprint arXiv:1802.08530. 2018 Feb 23.

---

### Official Review · Reviewer_FA7K · 2021-07-12

**Rating:** 6
**Confidence:** 4

**Summary:**

This paper proposes a new weight re-parametrization technique $S^3$ for training low-bit shift network. It claims that this new re-parametrization technique is weight-initialization insensitive and accuracy persistent compared with previous methods. In ImageNet, it achieves comparable top-1 accuracy with 3-bits, less than previous methods requires at 5-bit weight representation. Additionally, they also design two indices of weight dynamics to illustrate the mechanism of how $S^3$ works.

**Limitations And Societal Impact:**

Yes.

**Main Review:**

Originality: This paper proposes a novel weight re-parametrization method. More Related work on BNN or other sophisticated weight re-parametrization methods on low-bits networks should be considered.

Quality: There are serval doubts on the discussion on the "Gradient vanishing". In most cases on studies on low-bits networks, the backbone is mainly ResNet. With residual connection and batch norm, it might be impossible to cause VGP. It is hard to acknowledge the correspondence or relationship between VGP and the proposed $S^3$, especially in Binarized quantizer function since the weight representation is {0,1}.

For Equation 4, what exactly are these $\mathbf{W}$s representing? The decomposition mentioned in paper is unclear. What is the motivation for that? and any theoretical support for $w_{sparse}$ and $w_{sign}$. This can be really important. If the authors can illustrate somewhere in the main text, that would be appreciated.

One minor issue, it seems that the authors only use ImageNet dataset, therefore, it might be appropriate to remove descriptions such as, we evaluate on multiple datasets in Line 65. This can be misleading for readers.

Clarity: This paper is well-written, and the layout is good to read.

Significance: $S^3$ achieves lower bits than previous low-bits shift networks. However, the state-of-the-art is not sufficient. There are also Binary Neural Network which has even lower than the number reported in the paper, and in the same time, they achieve comparable results under the same setting [1]. More comparisons with BNN are encouraged here. It would be better if the author can demonstrate the applicability of training BNN.

[1] Haotong Qin, Ruihao Gong, Xianglong Liu, Mingzhu Shen, Ziran Wei, Fengwei Yu, and Jingkuan Song. Forward and backward information retention for accurate binary neural networks. In CVPR, 2020.


**Time Spent Reviewing:**

4

---

> ### Author Response · Authors · 2021-08-10
> **Author response**
>
> $\def\wshift{w_\mathrm{shift}}
> \def\wsparse{w_\mathrm{sparse}}
> \def\wsign{w_\mathrm{sign}}
> \def\wter{w_\mathrm{ter}}$
>
> Thank you for taking the time to review our paper and summarizing the main idea. Thank you for describing our paper as well-written and the layout is good to read and recognizing the originality of this work. We summarize your concerns as follows.
>
> #### **Q1:** ResNet is widely use in the study of low-bit networks. With residual connection + batchnorm, it might be impossible to cause VGP. Need more evidence to support VGP exist and S3 re-parameterization solve this issue.
>
> #### **A1:** In the camera-ready version, we will elaborate more on the vanishing gradient problem (VGP) description in section 3.1. Here we first clarify our VGP definition in its matrix multiplication form and then provide side evidence that it harms network performance even using residual connection and batch norm.
>
> **The VGP discussed in section 3.1 is the backprop gradient information loss due to one or more terms equal to zero in the backward propagation multiplication chain.** We are the first to point out that the sparse weight in a low-bit quantized network will lead to this VGP and propose s3 reparameterization as a remedy. The sparse derivative of ReLU leads to a very similar VGP in the ReLU-activated network, which is widely known as dying ReLU. As mentioned in several related works [1,2], dying ReLU is regarded as a special vanishing gradient problem.
>
> Firstly, let's use a simple example to clarify the VGP definition in this paper further. Consider a 2x2 matrix multiplication with ReLU activation from layer l to l+1,
>
> $\begin{pmatrix} x_{0}^{l+1}, \\\\  x_{1}^{l+1} \end{pmatrix} =  \begin{pmatrix}
>         \text{ReLU}(W_{00}^{l}x_{0}^{l} + W_{01}^{l}x_{1}^{l}),  \\\\
>         \text{ReLU}(W_{10}^{l}x_{0}^{l} + W_{11}^{l}x_{1}^{l})
>         \end{pmatrix}$
>
> The backward gradient update toward $x_{0}^{l}$ is computed as below,
>
> $\frac{\partial \text{Loss}}{\partial x_{0}^{l}}  = \frac{\partial \text{Loss}}{\partial x_{0}^{l+1}} \frac{\partial x_{0}^{l+1}}{\partial (W_{00}^{l}x_{0}^{l} + W_{01}^{l}x_{1}^{l})} W_{00}^{l} + \frac{\partial \text{Loss}}{\partial x_{1}^{l+1}}\frac{\partial x_{1}^{l+1}}{\partial (W_{10}^{l}x_{0}^{l} + W_{11}^{l}x_{1}^{l})} W_{10}^{l} $
>
> Where $ \frac{\partial x_{1}^{l+1}}{\partial (W_{10}^{l}x_{0}^{l} + W_{11}^{l}x_{1}^{l})}$ is the derivative of ReLU.
> $\frac{\\partial x_{1}^{l+1}}{\partial (W_{10}^{l}x_{0}^{l} + W_{11}^{l}x_{1}^{l})}=\left \\{ \begin{aligned}1 & & W_{10}^{l}x_{0}^{l} + W_{11}^{l}x_{1}^{l} > 0 \\\\ 0 & & W_{10}^{l}x_{0}^{l} + W_{11}^{l}x_{1}^{l} \leq 0 \end{aligned}\right. $
>
> Previous equations show that the gradient information $\frac{\partial \text{Loss}}{\partial x_{1}^{l+1}}$ propagating from $x_{1}^{l+1}$ to $x_{0}^{l}$ vanishes in two situations. (1) the corresponding derivative of ReLU is zero. (i.e. the neuron is not activated) (2) the corresponding weight value $W_{10}^{l}$ is zero.
>
> Dying ReLU issue refers to when (1) happens on all training examples. In this situation, the gradient will vanish forever, and a specific portion of weight will never receive gradient updates regardless of the feeding training examples. Up to 40\% of the network can be 'die' in ReLU-activated networks, and these weights cannot be sufficiently adjusted by the backprop gradient update signal [3] And (2) is the VGP discussed in section 3.1 of this work. We argue that (2) is much easier to occurs than (1) since the gradient vanishes once the corresponding weight value is zero, regardless of the input training example.
>
> Secondly, we provide side evidence to show **this type of VGP harms the neural network performance even with residual connection and batch norm**. Ramachandran et al. [4] benchmark 7 ReLU variants (LeakyReLU, PReLU, Softplus, ELU, SELU, GELU, Swish) on ResNet architecture. The positive half of these ReLU variants are similar to the vanilla ReLU. Still, they have a different design on the negative half to reduce the sparsity of the derivative of the activation function. The experimental results summarized in Table 4 show that 6 out of 7 ReLU variants out-perform vanilla ReLU on ResNet architecture. The ReLU variants with non-sparse derivative can improve the performance of neural network. We argue that the non-sparse weight tensors can provide a similar benefit under the same condition.
>
>
> #### **Q2:** The motivation of the sign-sparse decomposition is unclear, requires further theoretical support.
>
> #### **A2:** The sign-sparse decomposition design is motivated by two points,
> 1. The decomposed optimization space allows the ternary weight value to jump between -1 and 1 without across zero. This design avoid the weight sign freezing effect caused by the ternary quantizer. Our experiments in section 3.2 prove that S3 re-parameterization achieves its design goal, promoting weight sign flipping during training.
> 2. The independent sparse parameter in the decomposed space allows us to maximize the $L^{0}$ norm of weight tensor by applying a dense weight regularizer. Minimizing or maximizing the $L^{0}$ norm of weight tensor is a non-tractable problem in a continuous optimization space defined by the ternary quantizer.
>
>
> #### **Q3**  Table 1 and 2 should contain more multiplication-free networks as baselines.
>
> #### **A3**  More baselines [5, 6, 7, 8, 9, 10] are included in our first draft. They are removed from the final submission due to page limitations. We will re-allocate part of the ablation study to the appendix to save some space and bring these experimental results back to Tables 1/2 in the camera-ready version.  [11] will be added into Table 1 for comparison.
>
> Recap: we claim that $S^3$ reparameterization is the SOTA method for training shift neural networks. Binary networks only require 1bit weight but also has inferior performance compare to shift neural network.  Without changing network architecture (e.g. network widening), the binary networks can not achieve the same accuracy of full-precision models on ResNet 18/50 ImageNet experiments,  while shift neural networks can achieve.
>
> Reference:
>
> [1] Zheng Hu, Jiaojiao Zhang, and Yun Ge.  Handling vanishing gradient problem using artificial derivative.IEEE Access, 9:22371–22377, 2021.
>
> [2]Lu Lu, Yeonjong Shin, Yanhui Su, and George Em Karniadakis. Dying relu and initialization: Theory and numerical examples.arXiv preprint arXiv:1903.06733, 2019.
>
> [3] CS231N course note: https://cs231n.github.io/neural-networks-1/
>
> [4] Prajit Ramachandran, Barret Zoph, and Quoc V Le. Searching for activation functions.arXiv preprint arXiv:1710.05941, 2017.
>
> [5] Dongqing Zhang, Jiaolong Yang, Dongqiangzi Ye, and Gang Hua. Lq-nets: Learned quantization for highly accurate and compact deep neural networks. In Proceedings of the European conference on computer vision (ECCV), pages 365–382, 2018.
>
> [6] Yuhang Li, Xin Dong, and Wei Wang.  Additive powers-of-two quantization: An efficient non-uniform discretization for neural networks.arXiv preprint arXiv:1909.13144, 2019.
>
> [7] Xiaofan Lin, Cong Zhao, and Wei Pan. Towards accurate binary convolutional neural network.arXiv preprint arXiv:1711.11294, 2017.
>
> [8] Matthieu Courbariaux, Yoshua Bengio, and Jean-Pierre David. Binaryconnect: Training deep neural networks with binary weights during propagations.arXiv preprint arXiv:1511.00363, 2015.
>
> [9] Zhouhan Lin, Matthieu Courbariaux, Roland Memisevic, and Yoshua Bengio. Neural networks with few multiplications. arXiv preprint arXiv:1510.03009, 2015.
>
> [10] Oran Shayer, Dan Levi, and Ethan Fetaya. Learning discrete weights using the local reparameterization trick. arXiv preprint arXiv:1710.07739, 2017.
>
> [11] Haotong Qin, Ruihao Gong, Xianglong Liu, Mingzhu Shen, Ziran Wei, Fengwei Yu, and Jingkuan Song. Forward and backward information retention for accurate binary neural networks. In CVPR, 2020.

---

> > ### Author Response · Authors · 2021-08-25
> > **New table and 4bit result**
> >
> > **A3:** We update Table 1 with 4-bit result and add more recent related works for a better comparison. **S3-ShiftNN (4bit) reaches 70.47% Top1 accuracy on ImageNet.**
> >
> > |       Kernel      |    Methods    | Bit-width | Initialization | Top1 Accu. (%) | Top5 Accu. (%) |
> > |:-----------------:|:-------------:|:---------:|:--------------:|:--------------:|:--------------:|
> > |       Multi.      |      FP32     |     32    |     Random     |      69.6      |      89.2      |
> > |       Multi.      |     TTQ[4]    |     2     |   Pre-trained  |      66.6      |      87.2      |
> > | Sum of Sign Flips |   Lq-Net[1]   |     2     |     Random     |      68.0      |      88.0      |
> > | Sum of Sign Flips |   Lq-Net[1]   |     3     |     Random     |      69.3      |      88.8      |
> > | Sum of Sign Flips |   Lq-Net[1]   |     4     |     Random     |      70.0      |      89.1      |
> > | Sum of Xnor-Popcount |   ABC-Net[2]  |     5     |     Random     |      68.3      |      87.9      |
> > |     Sign Flip     |     BWN[5]    |     1     |     Random     |      60.8      |      83.0      |
> > |     Sign Flip     |    HWGQ[7]    |     1     |     Random     |      61.3      |      83.2      |
> > |     Sign Flip     |    BWHN[8]    |     1     |     Random     |      64.3      |      85.9      |
> > |     Sign Flip     |   IR-net[10]   |     1     |     Random     |      66.5      |      86.8      |
> > |     Sign Flip     |     TWN[6]    |     2     |     Random     |      61.8      |      84.2      |
> > |     Sign Flip     |   LR-net[3]   |     2     |     Random     |      63.5      |      84.8      |
> > |     Sign Flip     |   SQ-TWN[9]  |     2     |     Random     |      63.8      |      85.7      |
> > |     Sign Flip     |    INQ[12]    |     2     |   Pre-trained  |      66.02     |      87.13     |
> > |     Sign Flip     |      Ours     |     2     |     Random     |      66.37     |      87.18     |
> > | Shift + Sign Flip |    INQ[12]    |     3     |   Pre-trained  |      68.08     |      88.36     |
> > | Shift + Sign Flip |    INQ[12]    |     4     |   Pre-trained  |      68.89     |      89.01     |
> > | Shift + Sign Flip |    INQ[12]    |     5     |   Pre-trained  |      68.98     |      89.10     |
> > | Shift + Sign Flip | DeepShift[11] |     6     |     Random     |      65.63     |      86.33     |
> > | Shift + Sign Flip | DeepShift[11] |     6     |   Pre-trained  |      68.32     |      88.41     |
> > | Shift + Sign Flip |      Ours     |     3     |     Random     |      69.59     |      89.22     |
> > | Shift + Sign Flip |      Ours     |     4     |     Random     |      70.47     |      89.93     |
> >
> > 'Sum of Sign Flips' methods achieved similar accuracies to our method. However, they are more expensive to compute. For an n-bit weight, this kernel requires n sign-flip operations and (n-1) addition operations to compute the sum of sign-flips. Our method only requires one sign flip operation and one shift operation for n-bit weight. Note that the energy cost of addition operation and shift operation are similar on ASIC and FPGA [13].
> >
> > IR-net is a powerful method to train a 1bit network. The two optimization difficulties discussed in this paper: weight sign freezing and gradient vanishing, do not apply to a 1bit network. However, like many other 1bit networks, IR-net suffers from the accuracy drop caused by its low-bit, and it can not scale to a higher bit for better accuracy. When accuracy is necessary, we have to modify the network architecture to reduce the accuracy gap compare to the full-precision model. A commonly-used solution is network widening[14], which is expensive.  A 2x widening requires 2x activation memory occupation, 4x weight memory occupation, 4x storage size and 4x operations. It can quickly mitigate the performance benefit of 1-bit methods.
> >
> >
> > [1] Dongqing Zhang, Jiaolong Yang, Dongqiangzi Ye, and Gang Hua. Lq-nets: Learned quantization for highly accurate and compact deep neural networks. In Proceedings of the European conference on computer vision (ECCV), pages 365–382, 2018.
> >
> > [2] Xiaofan Lin, Cong Zhao, and Wei Pan. Towards accurate binary convolutional neural network.arXiv preprint arXiv:1711.11294, 2017.
> >
> > [3] Oran Shayer, Dan Levi, and Ethan Fetaya. Learning discrete weights using the local reparameterization trick. arXiv preprint arXiv:1710.07739, 2017.
> >
> > [4] Chenzhuo Zhu, Song Han, Huizi Mao, and William J Dally. Trained ternary quantization.arXiv preprint arXiv:1612.01064, 2016.
> >
> > [5] Mohammad Rastegari, Vicente Ordonez, Joseph Redmon, and Ali Farhadi. Xnor-net: Imagenet classification using binary convolutional neural networks. InEuropean conference on computer vision, pages 525–542. Springer, 2016.
> >
> > [6] Fengfu Li, Bo Zhang, and Bin Liu. Ternary weight networks.arXiv preprint arXiv:1605.04711, 2016.
> >
> > [7] Zhaowei Cai, Xiaodong He, Jian Sun, and Nuno Vasconcelos. Deep learning with low precision by half-wave gaussian quantization. InProceedings of the IEEE conference on computer vision and pattern recognition, pages 5918–5926, 2017.
> >
> > [8] Qinghao Hu, Peisong Wang, and Jian Cheng. From hashing to cnns: Training binary weight networks via hashing. In Thirty-Second AAAI conference on artificial intelligence, 2018.
> >
> > [9] Yinpeng Dong, Renkun Ni, Jianguo Li, Yurong Chen, Jun Zhu, and Hang Su. Learning accurate low-bit deep neural networks with stochastic quantization.arXiv preprint arXiv:1708.01001, 2017.
> >
> > [10] Haotong Qin, Ruihao Gong, Xianglong Liu, Mingzhu Shen, Ziran Wei, Fengwei Yu, and Jingkuan Song. Forward and backward information retention for accurate binary neural networks. In CVPR, 2020.
> >
> > [11] Mostafa Elhoushi, Zihao Chen, Farhan Shafiq, Ye Henry Tian, and Joey Yiwei Li. Deepshift: Towards multiplication-less neural networks.arXiv preprint arXiv:1905.13298, 2019.
> >
> > [12] Aojun Zhou, Anbang Yao, Yiwen Guo, Lin Xu, and Yurong Chen. Incremental network quantization: Towards lossless cnns with low-precision weights.arXiv preprint arXiv:1702.03044,1892017.
> >
> > [13] You, Haoran, et al. "Shiftaddnet: A hardware-inspired deep network." arXiv preprint arXiv:2010.12785 (2020).
> >
> > [14] McDonnell MD. Training wide residual networks for deployment using a single bit for each weight. arXiv preprint arXiv:1802.08530. 2018 Feb 23.

---

> ### Comment · Reviewer_FA7K · 2021-08-26
> **Comments**
>
> Thanks for the authors' feedbacks.
>
> After reading the rebuttal and updated materials, most of the concerns have been considered. I will raise my rating to 6.
>
> Best

---

> > ### Author Response · Authors · 2021-08-30
> > **Re: Comments**
> >
> > Thank you very much for reconsidering the rating of our submission. Please let us know if you have more concerns or if you need further clarification.
> >
> > Best,

---

### Official Review · Reviewer_CUjS · 2021-07-16

**Rating:** 8
**Confidence:** 3

**Summary:**

The paper presents the sign-sparsity-shift low-bit (i.e. 3-bit) reparametrization of weights for the multiplication-free networks, whose learning dynamics (as well as ImageNet results) closely resembles their full-precision counterparts.

**Limitations And Societal Impact:**

Yes.

**Main Review:**

Pros
+ The algorithm is efficient and easy to implement.
+ The empirical results are relatively impressive.
+ The writing is clear and easy to follow.

Cons
- The number of baselines being compared against in Table 1 & 2 [8, 12, 25] is simply too small, considering the heated competition in this area. As stated in Sec 2, there exist many papers about improving multiplication-free networks [23, 14, 24, 13] and discrete weights [5, 15, 18], which also should be compared against as much as possible (e.g. on a speed-accuracy chart) whether the authors believe they're too slow or not. Otherwise, the authors should provide a more detailed explanation regarding why they cannot be fairly compared against. In addition, [27] seems highly relevant but is missing in the paper.
- It's unclear why the shift network is parameterized recursively as St in Eq 6, which seems very inefficient in bit usage (when t is larger than 2, which wasn't but should also be tested in the paper) and strongly biases St towards 0 at initialization (if w's are all initialized symmetrically around 0, which is commonly the case). The alternative of using the staircase function (Eq 1) for the shift network should also be considered and tested in the paper.

[27] Ultra-Low Precision 4-bit Training of Deep Neural Networks, NeurIPS, 2020

Post rebuttal
- I'm satisfied with the authors' detailed feedback including comparison of their design of the shift network (Sec 4.2, Eq 6) against the more commonly used staircase function, additional ImageNet results of the 4-bit extension of their algorithm (which reasonably addressed my concern about its scalability) and additional, favorable comparison to a much wider range of prior work. I've upgraded my score to 8.

**Time Spent Reviewing:**

2

---

> ### Author Response · Authors · 2021-08-10
> **Author response**
>
> Thank you for taking the time to review our paper and summarizing the main idea. Thank you for describing our paper as well-written and recognizing easy implementation and impressive results as the main strengths of this work.
>
> However, you expressed valid concerns that we would like to address.
>
> #### **Q1:**  Table 1 and 2 should contain more multiplication-free networks as baselines or provide a better justification for no comparison. A speed-accuracy chart should be added. [27] could be a good baseline for comparison.
>
> #### **A1:** More baselines [1, 2, 3, 4, 5, 6] are included in our first draft. They are removed from the final submission due to page limitations. We will re-allocate part of the ablation study to the appendix to save some space and bring these experimental results back to Tables 1/2 in the camera-ready version.
>
> The speed-accuracy chart is a great suggestion but also a major additional step. It requires intensive engineering work to implement different algorithms in the hardware for comparison. Prior works such as [7] proposed FPGA implementation of shift neural network, and it achieves 4x saving on energy cost and 2.5x saving on FPGA resource compare to 8bit multiplication-based convolution.
>
> [27] focuses on training acceleration with a low-precision gradient—however, our work focus on inference acceleration with low-precision weight. Unfortunately, we can not add it to our table for comparison as [27] falls out of the scope of our work. We will refer to this work in the introduction section of our camera-ready version.
> [27] Ultra-Low Precision 4-bit Training of Deep Neural Networks, NeurIPS, 2020
>
> #### **Q2:** The alternative of using a staircase function for power-of-two scaling factors should consider as a comparison.
>
> #### **A2:** Our early experiments show the parameterization of power-of-two scaling factors with a staircase-like function leads to inferior performance. A more detailed ablation study is ongoing and will finish in a few days, and we will update the results on open-review and add them to the appendix of the camera-ready version.
>
>
>
> Reference:
>
> [1] Dongqing Zhang, Jiaolong Yang, Dongqiangzi Ye, and Gang Hua. Lq-nets: Learned quantization for highly accurate and compact deep neural networks. In Proceedings of the European conference on computer vision (ECCV), pages 365–382, 2018.
>
> [2] Yuhang Li, Xin Dong, and Wei Wang.  Additive powers-of-two quantization: An efficient non-uniform discretization for neural networks.arXiv preprint arXiv:1909.13144, 2019.
>
> [3] Xiaofan Lin, Cong Zhao, and Wei Pan. Towards accurate binary convolutional neural network.arXiv preprint arXiv:1711.11294, 2017.
>
> [4] Matthieu Courbariaux, Yoshua Bengio, and Jean-Pierre David. Binaryconnect: Training deep neural networks with binary weights during propagations.arXiv preprint arXiv:1511.00363, 2015.
>
> [5] Zhouhan Lin, Matthieu Courbariaux, Roland Memisevic, and Yoshua Bengio. Neural networks with few multiplications. arXiv preprint arXiv:1510.03009, 2015.
>
> [6] Oran Shayer, Dan Levi, and Ethan Fetaya. Learning discrete weights using the local reparameterization trick. arXiv preprint arXiv:1710.07739, 2017.
>
> [7] Denis A Gudovskiy and Luca Rigazio.  Shiftcnn: Generalized low-precision architecture for inference of convolutional neural networks.arXiv preprint arXiv:1706.02393, 2017.

---

> > ### Author Response · Authors · 2021-08-18
> > **Additional experiment results**
> >
> > **A2:**
> >
> > Here are the 3bit Shift NN experiment results of the alternative of using a staircase-like function for the shift network.
> >
> > In this experiment, we compare different parameterizations of the power-of-two scaling factors. Our proposed method is described in eq(6), and another is a staircase-like quantizer function. Besides different scaling factor parameterization, all other hyper-parameters and designs are the same as the experiments in Table3.
> >
> > The design of the staircase-like quantizer is following the general practice of quantization-aware training [8]. In 3bit Shift NN, the value of power-of-two scaling factors is limited to {0, 1, 2}. During the forward propagation, shift parameters rescale to the range of (-0.5, 2.5) based on their min and max, and then rounded. During the backward propagation, we use STE to estimate the gradient across the staircase-like quantizer function.
> >
> > | Parameterization of $2^p$ | ResNet18 ImageNet | $\alpha$=1e-2 | 1e-3 | 1e-4 | 1e-5 |
> > |---|---|---|---|---|---|
> > | Staircase-like function | Top1 / Top5 | 68.58 / 88.50 | 68.58 / 88.41 | 68.52 / 88.30 | 68.59 / 88.47 |
> > | Ours ($S^3$) | Top1 / Top5 | 69.78 / 89.21 | 69.85 / 89.24 | 69.84 / 89.23 | 69.82 / 89.23 |
> >
> > Our experiment results show that our proposed parameterization of the power-of-two scaling factor improves the shift NN's performance.
> >
> > [8] Krishnamoorthi R. Quantizing deep convolutional networks for efficient inference: A whitepaper. arXiv preprint arXiv:1806.08342. 2018.

---

> > > ### Comment · Reviewer_CUjS · 2021-08-19
> > > **re: Additional experiment results**
> > >
> > > Thank you very much for the additional results, which do help strengthen the paper. At the same time, I'm still curious why using the Krishnamoorthi staircase function is considerably worse than Eq 6, given that they should have the same representation power. Is the rescaling of the Krishnamoorthi staircase function better or worse than the clipping of Eq 1, which seems more commonly used as mentioned in L114? How's w of the Krishnamoorthi staircase function initialized? How's w_s1 and w_s2 (Eq 6, Fig 3) initialized? Most importantly, is initialization (distribution of w_shift at initialization) causing their learning dynamics to differ, thus their final accuracies, or is the staircase function really fundamentally worse for some reason?
> > >
> > > For 3-bit shift works, Eq 6 isn't really much more complicated than Eq 1 (clipping and rounding), but what about harder tasks that may require considerably more bits? Can the recursion of Eq 6 scale to handle such tasks better than Eq 1? As mentioned in my review, the scalability of the shift network is a concern in my opinion.
> > >
> > > Finally, the authors replied to 3 reviewers that more baselines were available in their first draft. Why not just provide comparisons to those baselines in the rebuttal? I think that's the most direct way to strengthen the paper.

---

> > > > ### Author Response · Authors · 2021-08-25
> > > > **re: re: Additional experiment results**
> > > >
> > > >
> > > > Thank you for your great comment and advice !
> > > >
> > > > **Q1:** I'm still curious why using the Krishnamoorthi staircase function (rescaling + rounding) is considerably worse than Eq 6, given that they should have the same representation power.
> > > >
> > > > **A1:** This is an interesting question and requires further investigation. A potential answer is an optimization difficulty similar to the weight sign freezing issue in the staircase function of the power-of-two scaling factors. As mentioned by reviewer geT6, our results can inspire follow-up theory research. We plan to generalize the weight sign freezing study to a higher-bit quantizer in a follow-up theory paper. We focus on analyzing the ternary quantizer in this paper since it is a key design in many training solutions of low-bit weight networks, such as TWN [11], TTQ [9], DeepShift [16] and INQ[17].
> > > >
> > > > **Q2:** Is the rescaling of the Krishnamoorthi staircase function better or worse than the clipping of Eq 1, which seems more commonly used as mentioned in L114?
> > > >
> > > > **A2:** Rescaling the continuous parameter is a more commonly-used quantizer design compare to clipping. From our own experience, rescaling before the staircase function helps to improve the accuracy, so we choose this solution.
> > > >
> > > > Eq 1. is a general form of quantizers (staircase function) that maps a continuous parameter to discrete values. Many different staircase function designs propose in the related works, and rescaling the continuous parameter is a popular choice.
> > > >
> > > > Here is a summary of the staircase function design in the related works.
> > > >
> > > > 1. Rescaling + Rounding to the nearest integer [8].
> > > > 2. Computing dynamic thresholds $t$ based on the max(abs(W)) [9, 11].
> > > > 3. Select the range of 2^p based on the max(abs(W)) [17].
> > > > 4. Standardize the continuous parameter and binarizing [15].
> > > > 5. Rounding to the nearest integer + clipping out of range value [16].
> > > >
> > > > Quantizers 2,3,4 are dynamically adjusted to fit the range of W, which is similar to rescaling. A uniform quantizer with value clipping is a less popular choice in the quantizer design.
> > > >
> > > > **Q3:** How's w of the Krishnamoorthi staircase function initialized? How's w_s1 and w_s2 (Eq 6, Fig 3) initialized? Most importantly, is initialization (distribution of w_shift at initialization) causing their learning dynamics to differ, thus their final accuracies, or is the staircase function really fundamentally worse for some reason?
> > > >
> > > >
> > > > **A3:** We use kaiming init for all variables, including the w of the staircase function. From our experience, the final accuracies of the S3 re-parameterization method are not sensitive to their initialization since the binary variables initialize to a range close to the threshold, and they are quickly oscillating around the threshold during training, so the initial values are less important. A potential explanation of the inferior performance of the staircase function discusses in A1.
> > > >
> > > > **Q4:** For 3-bit shift works, Eq 6 isn't really much more complicated than Eq 1 (clipping and rounding), but what about harder tasks that may require considerably more bits? Can the recursion of Eq 6 scale to handle such tasks better than Eq 1? As mentioned in my review, the scalability of the shift network is a concern in my opinion.
> > > >
> > > > **A4:** Very good question. We update the 4-bit result of our method in the new table below, and S3-Shift-4bit reaches 70.47% Top1 accuracy on ImageNet. This result shows our method is effective in the range of 2 to 4 bits. In this paper, we only focuses on the training of low-bit shift networks. We also think improving the representation efficiency for higher-bit training is an exciting question for follow-up works.
> > > >
> > > > **Q5:** Finally, the authors replied to 3 reviewers that more baselines were available in their first draft. Why not just provide comparisons to those baselines in the rebuttal? I think that's the most direct way to strengthen the paper.
> > > >
> > > > **A5:** Thank you for your good suggestion ! This is our updated Table 1 with more recently published works for a better comparison.
> > > >
> > > > |       Kernel      |    Methods    | Bit-width | Initialization | Top1 Accu. (%) | Top5 Accu. (%) |
> > > > |:-----------------:|:-------------:|:---------:|:--------------:|:--------------:|:--------------:|
> > > > |       Multi.      |      FP32     |     32    |     Random     |      69.6      |      89.2      |
> > > > |       Multi.      |     TTQ[9]    |     2     |   Pre-trained  |      66.6      |      87.2      |
> > > > | Sum of Sign Flips |   Lq-Net[1]   |     2     |     Random     |      68.0      |      88.0      |
> > > > | Sum of Sign Flips |   Lq-Net[1]   |     3     |     Random     |      69.3      |      88.8      |
> > > > | Sum of Sign Flips |   Lq-Net[1]   |     4     |     Random     |      70.0      |      89.1      |
> > > > | Sum of Xnor-Popcount |   ABC-Net[3]  |     5     |     Random     |      68.3      |      87.9      |
> > > > |     Sign Flip     |     BWN[10]    |     1     |     Random     |      60.8      |      83.0      |
> > > > |     Sign Flip     |    HWGQ[12]    |     1     |     Random     |      61.3      |      83.2      |
> > > > |     Sign Flip     |    BWHN[13]    |     1     |     Random     |      64.3      |      85.9      |
> > > > |     Sign Flip     |   IR-net[15]   |     1     |     Random     |      66.5      |      86.8      |
> > > > |     Sign Flip     |     TWN[11]    |     2     |     Random     |      61.8      |      84.2      |
> > > > |     Sign Flip     |   LR-net[6]   |     2     |     Random     |      63.5      |      84.8      |
> > > > |     Sign Flip     |   SQ-TWN[14]  |     2     |     Random     |      63.8      |      85.7      |
> > > > |     Sign Flip     |    INQ[17]    |     2     |   Pre-trained  |      66.02     |      87.13     |
> > > > |     Sign Flip     |      Ours     |     2     |     Random     |      66.37     |      87.18     |
> > > > | Shift + Sign Flip |    INQ[17]    |     3     |   Pre-trained  |      68.08     |      88.36     |
> > > > | Shift + Sign Flip |    INQ[17]    |     4     |   Pre-trained  |      68.89     |      89.01     |
> > > > | Shift + Sign Flip |    INQ[17]    |     5     |   Pre-trained  |      68.98     |      89.10     |
> > > > | Shift + Sign Flip | DeepShift[16] |     6     |     Random     |      65.63     |      86.33     |
> > > > | Shift + Sign Flip | DeepShift[16] |     6     |   Pre-trained  |      68.32     |      88.41     |
> > > > | Shift + Sign Flip |      Ours     |     3     |     Random     |      69.59     |      89.22     |
> > > > | Shift + Sign Flip |      Ours     |     4     |     Random     |      70.47     |      89.93     |
> > > >
> > > > 'Sum of Sign Flips' methods achieved similar accuracies to our method. However, they are more expensive to compute. For an n-bit weight, this kernel requires n sign-flip operations and (n-1) addition operations to compute the sum of sign-flips. Our method only requires one sign flip operation and one shift operation for n-bit weight. Note that the energy cost of addition operation and shift operation are similar on ASIC and FPGA [18].
> > > >
> > > > IR-net is a powerful method to train a 1bit network. The two optimization difficulties discussed in this paper: weight sign freezing and gradient vanishing, do not apply to a 1bit network. However, like many other 1bit networks, IR-net suffers from the accuracy drop caused by its low-bit, and it can not scale to a higher bit for better accuracy. When accuracy is necessary, we have to modify the network architecture to reduce the accuracy gap compare to the full-precision model. A commonly-used solution is network widening[19], which is expensive. A 2x widening requires 2x activation memory occupation, 4x weight memory occupation, 4x storage size and 4x operations. It can quickly mitigate the performance benefit of 1-bit methods.
> > > >
> > > > [9] Chenzhuo Zhu, Song Han, Huizi Mao, and William J Dally. Trained ternary quantization.arXiv preprint arXiv:1612.01064, 2016.
> > > >
> > > > [10] Mohammad Rastegari, Vicente Ordonez, Joseph Redmon, and Ali Farhadi. Xnor-net: Imagenet classification using binary convolutional neural networks. InEuropean conference on computer vision, pages 525–542. Springer, 2016.
> > > >
> > > > [11] Fengfu Li, Bo Zhang, and Bin Liu. Ternary weight networks.arXiv preprint arXiv:1605.04711, 2016.
> > > >
> > > > [12] Zhaowei Cai, Xiaodong He, Jian Sun, and Nuno Vasconcelos. Deep learning with low precision by half-wave gaussian quantization. InProceedings of the IEEE conference on computer vision and pattern recognition, pages 5918–5926, 2017.
> > > >
> > > > [13] Qinghao Hu, Peisong Wang, and Jian Cheng. From hashing to cnns: Training binary weight networks via hashing. InThirty-Second AAAI conference on artificial intelligence, 2018.
> > > >
> > > > [14] Yinpeng Dong, Renkun Ni, Jianguo Li, Yurong Chen, Jun Zhu, and Hang Su. Learning accurate low-bit deep neural networks with stochastic quantization.arXiv preprint arXiv:1708.01001, 2017.
> > > >
> > > > [15] Haotong Qin, Ruihao Gong, Xianglong Liu, Mingzhu Shen, Ziran Wei, Fengwei Yu, and Jingkuan Song. Forward and backward information retention for accurate binary neural networks. In CVPR, 2020.
> > > >
> > > > [16] Mostafa Elhoushi, Zihao Chen, Farhan Shafiq, Ye Henry Tian, and Joey Yiwei Li. Deepshift: Towards multiplication-less neural networks.arXiv preprint arXiv:1905.13298, 2019.
> > > >
> > > > [17] Aojun Zhou, Anbang Yao, Yiwen Guo, Lin Xu, and Yurong Chen. Incremental network quantization: Towards lossless cnns with low-precision weights.arXiv preprint arXiv:1702.03044,1892017.
> > > >
> > > > [18] You, Haoran, et al. "Shiftaddnet: A hardware-inspired deep network." arXiv preprint arXiv:2010.12785 (2020).
> > > >
> > > > [19] McDonnell MD. Training wide residual networks for deployment using a single bit for each weight. arXiv preprint arXiv:1802.08530. 2018 Feb 23.

---

### Decision · Program_Chairs · 2021-09-27

**Decision:**

Accept (Poster)

**Comment:**

Following the authors' response, this paper had two very positive reviewers (8), one slightly positive reviewer (6), and one very negative reviewer (2).

The main contribution of this paper, that excited the positive reviewers, was it showed for the first time it is possible to train from scratch with shift based quantization on ImageNet, and at a very low precision (3bit). This surprising observation that the suggested re-parameterization + "densifying regularization" are needed is insightful and should be helpful for developing other methods.

The very negative reviewer was mainly concerned with the re-parameterization, and suggested many weights should get "stuck" during optimization. Though the reviewer argument initially had some errors in the phase space diagram, we had a discussion whether the correct phase space diagram could should have such issues. However, during the discussion, I became convinced this is not a critical issue, because of the toy examples below (found in the discussion) by the most positive reviewer, in which we actually converge to the correct solution. Interestingly, this convergence seems to work be because of the "densifying regularization", and I encourage the authors to verify and discuss this. The results added by the authors during the review process are useful for this.

I would ask also the authors to polish the writing to help readability and impact of this paper. I tend to agree with the negative reviewer that some parts of this work are imprecise and confusing (and so did even the most positive reviewer).

**** Example ****

Suppose the training loss is $\frac12 (w_{ter} - 0)^2$ then $\frac{dL}{dw_{ter}} = g = w_{ter}= H(x)(2 H(y) - 1)$

Then, the STE gradient of the regularized loss L is

$\frac{dL}{dx} = ((2 H(y) - 1)  H(x) )  (2 H(y) - 1)    - \alpha I_{x<0}  = H(x) - \alpha H(-x) $ (i.e., 1 for x>0, -$\alpha$ for x<0)

, where $H$ is the heaviside function. Therefore the step $-\frac{dL}{dx}$ will always drive $x$ toward 0, and if $\alpha$ is small, we will have $x<0$ for most iterations.

Another one example:
$L = \frac12 (w_{ter} - 1)^2$, $g = w_{ter} = H(x)(2 H(y) - 1) - 1  $

$\frac{dL}{dx} = ((2 H(y) - 1)  H(x) - 1)  (2 H(y) - 1)     - \alpha I_{x<0}  = H(x) - (2 H(y) - 1)  - \alpha H(-x) $

$\frac{dL}{dy} = 2 H(x) g = 2 H(x) (( H(x) (2 H(y) - 1)) -1)  = 2 H(x) ((2 H(y) - 1)  - 1) = 4 H(x) (H(y) - 1) $

so, the step $-\frac{dL}{dy} = -4H(x)(H(y) - 1) $ will be non zero only if $x>0$ and $y < 0$. If $x>0$ it will eventually become positive as required.

If, on the other hand, $y<0$, the step $-\frac{dL}{dx} =   -H(x) - 1 + \alpha H(-x)  $  so it will move down while x<0, and, indeed, stay negative. Therefore, (-1,-1) won't shift to (1,1) in this setup.